# Provably Optimal Memory Capacity for Modern Hopfield Models: Transformer-Compatible Dense Associative Memories as Spherical Codes

**Jerry Yao-Chieh Hu**[*†‡]    **Dennis Wu**[*‡]    **Han Liu**[†‡§]

[†]Center for Foundation Models and Generative AI, [‡]Department of Computer Science, [§]Department of Statistics and Data Science, Northwestern University, Evanston, IL 60208, USA
{jhu,hibb}@u.northwestern.edu, hanliu@northwestern.edu

## Abstract

We study the optimal memorization capacity of modern Hopfield models and Kernelized Hopfield Models (KHMs), a transformer-compatible class of Dense Associative Memories. We present a tight analysis by establishing a connection between the memory configuration of KHMs and spherical codes from information theory. Specifically, we treat the stored memory set as a specialized spherical code. This enables us to cast the memorization problem in KHMs into a point arrangement problem on a hypersphere. We show that the optimal capacity of KHMs occurs when the feature space allows memories to form an optimal spherical code. This unique perspective leads to: (i) An analysis of how KHMs achieve optimal memory capacity, and identify corresponding necessary conditions. Importantly, we establish an upper capacity bound that matches the well-known exponential lower bound in the literature. This provides the first tight and optimal asymptotic memory capacity for modern Hopfield models. (ii) A sub-linear time algorithm U-Hop+ to reach KHMs' optimal capacity. (iii) An analysis of the scaling behavior of the required feature dimension relative to the number of stored memories. These efforts improve both the retrieval capability of KHMs and the representation learning of corresponding transformers. Experimentally, we provide thorough numerical results to back up theoretical findings.

## 1  Introduction

We study the optimal memorization capacity of Kernelized modern Hopfield Models (KHMs) [Wu et al., 2024a], propose a sublinear-time algorithm to achieve it, and analyze parameter selection for these models. KHMs belong to a class of *transformer-compatible* Dense Associative Memory [Krotov and Hopfield, 2021, 2016] known as Modern Hopfield Models (MHMs) [Wu et al., 2024a,b, Hu et al., 2024a, 2023, Ramsauer et al., 2020]. The defining characteristics of these models include their super-linear memory capacity and strong connection to transformer attention mechanisms [Vaswani et al., 2017]. The former makes them interesting models for associative memory, and the latter makes them versatile transformer-compatible backbones with diverse empirical successes [Burns, 2024, Burns and Fukai, 2023, Hu et al., 2024a,c, Xu et al., 2024, Wu et al., 2024a,b, Hoover et al., 2024b, Seidl et al., 2022, Fürst et al., 2022]. However, one major limitation of MHMs is their reliance on the quality of memory distribution for effective pattern storage and retrieval [Wu et al., 2024a, Sec. 1].

Studying this limitation in these models is fundamental and of practical importance. One one hand, it prevents MHMs from functioning as full-fledged content-addressable memory models. On the other hand, it implies that the representation learning ability of current transformer attention [Vaswani et al., 2017] is suboptimal [Wu et al., 2024a, Thm. 3.1]. Addressing this issue benefits both computational

---

[*]Equal contribution. Code is available at GitHub. Latest version is on arXiv.

38th Conference on Neural Information Processing Systems (NeurIPS 2024).

neuroscience and large foundation model research [Bietti et al., 2024, Krotov, 2023, Kozachkov et al., 2023, Cabannes et al., 2023, Hoover et al., 2023]. Kernelized modern Hopfield Models (KHMs) [Wu et al., 2024a] alleviate this issue by storing memories in the kernelized feature space. A key advantage of KHMs is their ability to reposition memories in the feature space, resulting in larger storage capacity. However, despite strong empirical performance, their capacity still lacks an optimal guarantee [Wu et al., 2024a, Sec. 5]. In this work, we close this gap by establishing the optimality of KHMs' memory capacity and presenting a sublinear-time algorithm to achieve it.

Let $\Xi := [\xi_1, \cdots, \xi_M] \in \mathbb{R}^{d \times M}$ be a set of memory patterns where each column (indexed by $\mu \in [M]$) represents a memory $\xi_\mu \in \mathbb{R}^d$, and let $x \in \mathbb{R}^d$ be the input query. The Hopfield models [Hopfield, 1982] are energy-based associative memory models, which store memories on the local minima of their energy functions. They retrieve a pattern by iteratively updating the query $x^t \mapsto x^{t+1}$ with its update rule $\mathcal{T}(x^t)$, for some $t \in \{0, 1, ...\}$. This update rule converge to a fixed point $x^\star$, defined by $x^\star = \mathcal{T}(x^\star)$. $x^\star$ is the retrieved pattern[2] based on initial query $x^0$.

Explicitly, iteratively updating $x$ with $\mathcal{T}$ is defined as a process of minimization to an energy function $E(x)$. For example, the Modern Hopfield Model [Ramsauer et al., 2020] has the energy function:

$$E(x) = \frac{1}{2} \langle x, x \rangle + \mathrm{lse}\left(\beta, \Xi^\mathsf{T} x\right), \tag{1.1}$$

where $\mathrm{lse}(\beta, z) := \log\left(\sum_{\mu=1}^M \exp\{\beta z_\mu\}\right)$, with some $\beta > 0$. With the Concave-Convex Procedure (CCCP) [Yuille and Rangarajan, 2001], (1.1) is monotonically decreased by an iterative update rule

$$x^{t+1} \leftarrow \mathcal{T}(x^t) = \Xi \cdot \mathrm{Softmax}(\beta \Xi^\mathsf{T} x^t). \tag{1.2}$$

This design boosts MHMs to store *exponentially* (in pattern dimension $d$) many memories compared to the *linear* capacity of the classic Hopfield model [Hopfield, 1982]. It also provides a model-based interpretation of the transformer attention mechanism [Wu et al., 2024a,b, Hu et al., 2024a,b, 2023, Ramsauer et al., 2020]. However, their retrieval accuracy and memory capacity hinge on the quality of the stored memory set [Wu et al., 2024a, Sec. 1], and hence are suboptimal in most scenarios.

To be concrete, for retrieving the $\mu$-th memory ($\mu \in [M]$), the retrieval error of MHM is exponentially suppressed by the pattern separation: $\Delta_\mu := \langle \xi_\mu, \xi_\mu \rangle - \max_{\nu, \nu \neq \mu} \langle \xi_\nu, \xi_\mu \rangle$ ([Wu et al., 2024a, Eqn. 1.3] or [Hu et al., 2023, Eqn. 2.7]). This $\Delta_\mu$-dependence in MHM retrieval accuracy also manifests the $\Delta_\mu$-dependence in memory capacity. To see this, recall that the standard memory capacity is a high-probability bound based on thresholding the separation $\Delta_\mu$ for each pattern $\mu \in [M]$ to determine storage and retrieval (Section 2.1). Explicitly, storing a pattern requires its separation to exceed a threshold that decreases with the *minimal* separation: $\Delta_{\min} := \min_{\mu \in [M]} \Delta_\mu$. Namely, a larger $\Delta_{\min}$ leads to larger capacity (Appendix B.1). Thus, the capacity depends on $\Delta_{\min}$. Yet, $\Delta_{\min}$ depends on the stored memories $\Xi$. This $\Xi$-dependence makes the capacity suboptimal.

Wu et al. [2024a] relax such limitation by introducing a kernel as a learnable similarity measure, using stored memory patterns as training data to enhance memory capacity. Specifically, they propose the Kernelized Modern Hopfield Model (KHM) defined by following update rule and energy function:

$$x^{t+1} \leftarrow \mathcal{T}_\Phi(x^t) := \Xi \cdot \mathrm{Softmax}\left(\beta \mathcal{K}(\Xi, x)\right), \quad E_\mathcal{K}(x) = \frac{1}{2}\mathcal{K}(x, x) + \mathrm{lse}\left(\beta, \mathcal{K}(\Xi, x)\right), \tag{1.3}$$

where the kernel $\mathcal{K}(\cdot, \cdot) := \langle \Phi(\cdot), \Phi(\cdot) \rangle : \mathbb{R}^d \times \mathbb{R}^d \to \mathbb{R}$ is associated with a learnable feature map $\Phi : \mathbb{R}^d \to \mathbb{R}^{D_\Phi}$. Here, $\mathcal{K}(\cdot, \cdot)$ acts column-wise on matrix: $\mathcal{K}(\Xi, x) = [\{\mathcal{K}(\xi_\mu, x)\}_{\mu=1}^M] = [\{\langle \Phi(\xi_\mu), \Phi(x) \rangle\}_{\mu=1}^M] \in \mathbb{R}^M$. Importantly, KHMs shift the dependency on $\Delta_{\min}$ to $\Delta_{\min}^\Phi$, with

$$\Delta_{\min}^\Phi := \min_{\mu \in [M]} \Delta_\mu^\Phi, \quad \text{where} \quad \Delta_\mu^\Phi := \mathcal{K}\left(\xi_\mu, \xi_\mu\right) - \max_{\nu, \nu \neq \mu} \mathcal{K}\left(\xi_\nu, \xi_\mu\right).$$

Notably, $\Delta_{\min}^\Phi$ is learnable and parameterized via $\Phi$. Wu et al. [2024a] point out that with $\Phi(x) = Wx$, where $W \in \mathbb{R}^{d \times D_\Phi}$, finding a suitable $\Phi$ that maximizes $\Delta_{\min}^\Phi$ benefits memory storage. This construction of $\Phi$ preserves key MHM properties, such as accurate [Wu et al., 2024a, Lemma 2.1] and consistent [Wu et al., 2024a, Thm 2.1] retrieval. However, direct maximization of $\Delta_{\min}^\Phi$ is

---

[2] $x^\star$ corresponds either to one of the memories or to a *fuzzy* combination of them. Please see [Ramsauer et al., 2020, Sec. A.1.5] for an informal discussion, [Wu et al., 2024a, Sec. 1] for further discussion, and [Santos et al., 2024, Wu et al., 2024a] for conditions of exact retrieval through sparsity [Wu et al., 2024b, Hu et al., 2023].

challenging due to its max-min nature. To circumvent, Wu et al. [2024a] propose a surrogate loss to maximize $\Delta_\mu^\Phi$ *on average* [Wu et al., 2024a, Def. 2.2]. As a result, their approach achieves strong empirical results in memory retrieval for MHMs and supervised learning for transformer models.

Nevertheless, maximizing $\Delta_\mu^\Phi$ on average, rather than $\Delta_{\min}^\Phi$, raises questions about how their surrogate loss benefits memory storage. Moreover, the impact of $\Delta_{\min}^\Phi$ on memory capacity lacks a clear analytical characterization, and no theoretical analysis confirms whether maximizing $\Delta_{\min}^\Phi$ leads to optimal memorization capacity.

In this paper, we address these questions from the perspective of (optimal) spherical codes from information thoery [Delsarte et al., 1991]. A spherical code is a set of points (vectors) distributed on the surface of a hypersphere, and an optimal spherical code is when the minimum angular distance between any two points is maximized. In other words, optimal spherical codes aim to spread the points as evenly as possible over the surface of the sphere. This aligns with the intuition behind KHM — increasing average separation between stored memories improves memory capacity. Therefore, we treat the stored memory pattern set as a spherical code (Definition 2.3), and require this spherical code to satisfy the *well-separation condition* [Hu et al., 2023, Thm 3.1]. We term this spherical code as *memory code*. Surprisingly, this unique connection enables a tight analysis on KHMs' capacity.

**Contributions.** Through the memory code perspective, this work makes three main contributions:

- **Provably Optimal Capacity.** We study the optimal memory capacity of KHMs and identify the conditions necessary to achieve it. Specifically, we derive a provably tight and optimal capacity by matching the well-known exponential lower bound for the memory capacity of MHMs [Wu et al., 2024a,b, Hu et al., 2023, Ramsauer et al., 2020] with an upper bound in the low-temperature region. Notably, we establish this tight bound by showing that KHMs store the most memories when the memory set forms an optimal spherical code (Lemma 2.2). This result suggests a tight exponential scaling of memory capacity with the pattern dimension $D_\Phi$ (Proposition 2.1).

- **Fast Algorithm.** We introduce an algorithm, U-Hop+, that achieves the optimal capacity of KHM in sublinear time. Theoretically, we show that, as temperature approaches zero, U-Hop+ finds the optimal feature map for maximal KHM capacity (Theorem 3.1). This result bridges our theoretical findings with practical applications and explains the empirical successes of [Wu et al., 2024a].

- **Numerical Validation.** Our experiments validate our theoretical analysis. We observe that (i) U-Hop+ creates distinct low-energy regions for each memory pattern, addressing the memory confusion problem in MHMs [Wu et al., 2024a, Krotov and Hopfield, 2016]; (ii) U-Hop+ significantly reduces metastable states on both MNIST and synthetic datasets, indicating larger memory capacity; (iii) with U-Hop+, the KHMs update rule converges to fixed points faster.

**Organization.** Section 1 presents a brief review of MHMs and KHMs. Appendix A includes related work discussions. Section 2 presents our main results. Specifically, Section 2.1 presents a memory capacity lower bound for KHMs, Section 2.2 presents the optimal capacity bound based on the notation of memory code. Section 3.1 presents a sublinear time algorithm to search for the optimal $\Phi$. Section 3.2 discusses the relationship between $\Phi$ and $M$. Section 4 includes numerical experiments.

**Notations.** Lower case letters denote (column) vectors and upper case letters denote matrices. We write $\langle a, b \rangle := a^\mathsf{T} b$ as the inner product for vectors $a, b \in \mathbb{R}^d$. The index set $\{1, ..., I\}$ is denoted by $[I]$, where $I \in \mathbb{N}^+$. The spectral norm is denoted by $\|\cdot\|_2$ which is equivalent to the $\ell_2$-norm when applied to a vector. We denote the memory patterns (keys) by $\xi \in \mathbb{R}^d$ and the query pattern by $x \in \mathbb{R}^d$, and $\Xi := [\xi_1, ..., \xi_M] \in \mathbb{R}^{d \times M}$ as shorthand for stored memory patterns $\{\xi_\mu\}_{\mu \in [M]}$. Throughout this work, we use $\Xi$ interchangeably to refer to either a $d \times M$ matrix or a set of $M$ $d$-dimensional memory pattern vectors.

## 2 Main Theory

We provide a theoretical analysis on the optimal memory capacity of KHMs. First, we begin by comparing the memory capacity between MHM and KHM using the standard high-probability lower bound [Hu et al., 2023, Ramsauer et al., 2020]. Then, we present a spherical code perspective as a framework for depicting the optimal memory capacity of both MHMs and KHMs. In our analysis, we make the following pattern normalization assumption on memory patterns:[3]

---

[3]It is a common assumption adopted in [Santos et al., 2024, Wu et al., 2024a]. A justification is the connection to transformer-attention. The modern Hopfield update rule (a.k.a. retrieval dynamics) is constantly compared to

**Assumption 1.** We assume memory patterns $\|\xi_\mu\| = 1$ in the rest of our paper.

## 2.1 High-Probability Capacity Lower Bound

We start by showing the memory capacity of KHM using the standard capacity lower bound introduced by Ramsauer et al. [2020]. This provides a direct comparison between KHMs and previous works. The definition of the generalized fixed point [Sriperumbudur and Lanckriet, 2009] is

**Definition 2.1** (Generalized Fixed Point [Sriperumbudur and Lanckriet, 2009]). We say a set $S \subseteq \mathbb{R}^d$ is a *generalized fixed point* w.r.t. $\mathcal{T}$ if $\mathcal{T}(y) \in S$ for every $y \in S$.

**Remark 1.** In contrast to Definition 2.1, a fixed point of $\mathcal{T}$ is a point $y$ satisfying $\mathcal{T}(y) = y$.

Let $S_\mu^\Phi$ be a ball with radius $R_\Phi$[4] centered at every memory pattern in the feature space $\Phi(\xi_\mu)$:

$$S_\mu^\Phi = \{y \mid \|\Phi(\xi_\mu) - y\| \leq R_\Phi\}, \quad \text{where} \quad R_\Phi := \frac{1}{2} \min_{\substack{\mu,\nu \in [M] \\ \mu \neq \nu}} \|\Phi(\xi_\mu) - \Phi(\xi_\nu)\|. \tag{2.1}$$

Following [Wu et al., 2024a], we define the memory storage and retrieval as:

**Definition 2.2** (Pattern Storage and Retrieval). We say a memory pattern $\xi_\mu$ is *stored* if $S_\mu^\Phi$ is a generalized fixed point of $\mathcal{T}$, and there exists a fixed point $x_\mu^\star \in S_\mu^\Phi$. A memory pattern $\xi_\mu$ gets $\epsilon$-*retrieved* by $\mathcal{T}$ with an input query $x$ if $\|\mathcal{T}(x) - \xi_\mu\| \leq \epsilon$.

This definition is compatible with both KHMs and MHMs (with identity feature map). Under Definition 2.2, KHM's memory capacity is lower bounded by the following lemma.

**Lemma 2.1** (Memory Capacity of KHM). Let $1 - p$ be the probability of successfully storing and retrieving a pattern. Assuming the patterns are normalized, the number of patterns $M_\Phi$ that can be stored and retrieved by the KHM, following the update rule (1.3), is lower-bounded by:

$$M_\Phi \geq \sqrt{p} C^{(D_\Phi - 1)/4},$$

where $C$ is the solution to $C = b/(W_0(\exp\{a + \ln b\}))$, with $W_0(\cdot)$ being the principal branch of Lambert $W$ function, $a := (4/(D_\Phi - 1))(\ln((2\sqrt{p}-2)/R_\Phi) + 1)$ and $b := 4\beta/(5(D_\Phi - 1))$. For comparison, $M_\Phi$ reduces to MHM's capacity lower bound by setting $\Phi = I_d$, with $D_\Phi = d$.

*Proof.* Our proof follows [Hu et al., 2023, Wu et al., 2024b]. See Appendix C.1 for a proof. □

With a fixed $D_\Phi$, the highest lower bound of Lemma 2.1 corresponds to specific a $\Phi$ that maximizes $R_\Phi$. This provides an intuitive insight on the design of separation loss [Wu et al., 2024a, Definition 2.2] for kernel learning in [Wu et al., 2024a, Algorithm 1]. With an additional feature space, KHM has an exponential memory capacity in $D_\Phi$ that does not depend on $d$. When $D_\Phi = d$, KHMs obtain a tighter lower bound than MHMs if $R_\Phi > R$. This bound connects the storage capacities of KHMs and MHMs, showing that their capacities scale exponentially with respect to $D_\Phi$ and $d$.

## 2.2 Memory Code: Memories as Spherical Code

There are two aspects the lower bound in Lemma 2.1 does not address: the maximal capacity of KHMs and the flexibility of choosing different $\Phi$ in KHMs. Therefore, we present a new framework using spherical codes to take the above perspectives into consideration for further analysis. We begin by introducing the concepts of spherical code and optimal spherical code.

**Definition 2.3** (Spherical Code). A $d$-dimensional spherical code on the unit sphere $\mathbb{S}^{d-1}$ is a finite set $\mathcal{C}_N = \{c_1, ..., c_N\}$ of $\mathbb{S}^{d-1}$ with $N$ points, where $c_i \in \mathbb{R}^d$ for $i \in [N]$ and $|\mathcal{C}_N| = N$.

**Definition 2.4** (Minimal Separation). The minimal separation $\rho(\mathcal{C}_N)$ of a spherical code $\mathcal{C}_N$ is the maximal inner product between two distinct points in $\mathcal{C}_N$:

$$\rho(\mathcal{C}_N) = \max_{c_i, c_j \in \mathcal{C}_N} \langle c_i, c_j \rangle, \quad \text{for every } i \neq j.$$

---

attention-mechanism and the input query and memory set correspond to $Q$, $K$ for attention. As LayerNorm is a commonly used strategy in attention layers, this setup can also be seen in real-world scenarios.

[4]By definition, neighborhoods do not overlap: $S_\mu^\Phi \cap S_\nu^\Phi = \emptyset$ for $\nu \neq \mu$.

**Definition 2.5** (Optimal Spherical Code). Let $\mathcal{C}_N = \{c_1, \ldots, c_N\} \subseteq \mathbb{S}^{d-1}$ be a $d$-dimensional spherical code with $N$ points. An optimal spherical code $\mathcal{C}_N^\star$ minimizes the maximal pairwise inner product, which corresponds to maximizing the minimal separation between points in the code. Formally, the optimal spherical code $\mathcal{C}_N^\star$ is defined as:

$$\mathcal{C}_N^\star = \operatorname*{argmin}_{\mathcal{C}_N \subset \mathbb{S}^{d-1}} \max_{i \neq j} \langle c_i, c_j \rangle, \quad \text{for } i, j \in [N].$$

The minimal separation of the optimal spherical code is denoted as $\rho^\star$.[a]

---

[a]The optimal arrangement of most spherical codes is unknown, except for specific pairs of $(d, N)$. A list of known optimal arrangements and minimal separations can be found at http://neilsloane.com/packings/ and in [Conway and Sloane, 2013].

Next, we recall the function class $\mathcal{H}$ of the linear feature map introduced by Wu et al. [2024a]:

**Definition 2.6.** The function class $\mathcal{H}$ consists of linear maps that satisfy the following properties:
1. For all $\Phi \in \mathcal{H}$, $\Phi : \mathbb{S}^{d-1} \to \mathbb{S}^{D_\Phi - 1}$ is a linear map defined by a matrix $W \in \mathbb{R}^{d \times D_\Phi}$.
2. The matrix $W$ has full column rank.
3. When applying $\Phi$ to different inputs:
   - For a vector $\xi \in \mathbb{R}^d$, $\Phi(\xi) = W^\mathsf{T} \xi \in \mathbb{R}^{D_\Phi}$.
   - For a matrix $\Xi \in \mathbb{R}^{d \times M}$, $\Phi(\Xi) = (\Phi(\xi_1), \ldots, \Phi(\xi_M)) \in \mathbb{R}^{D_\Phi \times M}$.
   - For a set of vectors $\mathcal{V} = \{v_1, \ldots, v_N\}$, $\Phi(\mathcal{V}) = \{\Phi(v_1), \ldots, \Phi(v_N)\}$ with $|\Phi(\mathcal{V})| = N$.

Definition 2.6 ensures KHMs with feature map $\Phi(\cdot) \in \mathcal{H}$ satisfying the defining characteristics of MHMs: accurate [Wu et al., 2024a, Lemma 2.1] and consistent [Wu et al., 2024a, Thm 2.1] retrieval according to Definition 2.2.[5] Now, we combine the concept of spherical code and memory storage.

**Definition 2.7** (Kernelized Well-Separation Condition [Wu et al., 2024a,b, Hu et al., 2023, Ramsauer et al., 2020]). Given a set of kernelized memory patterns $\Phi(\Xi) = \{\Phi(\xi_\mu)\}_{\mu=1}^M \subseteq \mathbb{S}^{D_\Phi - 1}$, the kernelized memory pattern $\Phi(\xi_\mu)$ satisfies the well-separation condition if the following holds:

$$\Delta_\mu^\Phi \geq \frac{1}{\beta} \ln\left(\frac{2(M-1)}{R_\Phi}\right), \tag{2.2}$$

where the inverse temperature $\beta$ is given by (1.3) and $R_\Phi$ is defined by (2.1).

The inequality (2.2) is a necessary condition for the $\mu$-th memory to have a well-defined attractor basin. Hence, the more memories satisfying (2.2) the greater the memory capacity of the model.

**Definition 2.8** (Memory Code). Let $M \in \mathbb{N}_+$, $\beta > 0$, $D_\Phi > 1$ and $\Phi \in \mathcal{H}$. For any finite set $\Phi(\Xi) = \{\Phi(\xi_\mu)\}_{\mu=1}^M \subseteq \mathbb{S}^{D_\Phi - 1}$, we say the set $\Phi(\Xi)$ is a memory code if all points in $\Phi(\Xi)$ satisfies (2.2). Further, we denote $\Lambda_{D_\Phi}$ as the set of all memory codes in $\mathbb{S}^{D_\Phi - 1}$, including all possible $\Xi$, $\Phi$.

Notably, $\Lambda$ includes all the possible pattern sets $\{\Phi(\Xi)\}$ that are able to be stored and retrieved by kernelized Hopfield models and modern Hopfield models. Naturally, the optimal memory capacity is the size of the largest memory code in $\mathbb{S}^{D_\Phi - 1}$. This leads to our next definition:

**Definition 2.9** (Optimal Memory Capacity). For $D_\Phi > 1$ and $\Phi \in \mathcal{H}$, the optimal capacity $M^\star$ is the cardinality of the largest memory code in $\Lambda_{D_\Phi}$, i.e., $M^\star := \max_{\Phi(\Xi) \in \Lambda} |\Phi(\Xi)|$ for all possible $\Xi$, $\Phi$.

Definition 2.9 specifies the largest possible memory code in $\Lambda_{D_\Phi}$ for a given $D_\Phi$. Let $\widetilde{\Xi}$ denote the memory set associated with $M^\star$, such that $\|\widetilde{\Xi}\| = M^\star$. To store all patterns in $\widetilde{\Xi}$, we need to find a suitable feature map $\widetilde{\Phi}$ such that $\widetilde{\Phi}(\widetilde{\Xi})$ is a valid memory code.

Following this definition, we present the next lemma and proposition on optimal memory capacity.

**Lemma 2.2** (Capacity of Optimal Spherical Code). Given a fixed $D_\Phi > 1$, and its corresponding $M^\star$, if an optimal code $\mathcal{C}_{\text{opt}}$ is in $\mathbb{S}^{D_\Phi - 1}$ and has size $M^\star$, then $\mathcal{C}_{\text{opt}} \in \Lambda_{D_\Phi}$.

---

[5]Note that, Definition 2.6 is sufficient but not necessary. It is possible to find a different $\mathcal{H}'$ such that KHMs with $\Phi \in \mathcal{H}'$ achieves desiring characteristics as [Wu et al., 2024a, Lemma 2.1] and [Wu et al., 2024a, Thm 2.1].

**Proposition 2.1** (Optimal Memory Capacity). Following Lemma 2.2, we have

$$M^\star \asymp c^{D_\Phi},$$

for some $c > 1$. Here $\asymp$ indicates matching upper and lower bounds up to constant factors.

*Proof Sketch.* We proof Lemma 2.2 by showing that the model capacity is a increasing function w.r.t. the minimal separation value. For Proposition 2.1, we utilize the upper bound in [Kabatiansky and Levenshtein, 1978] and lower bound in [Wyner, 1965, Shannon, 1959, Chabauty, 1953] to bound the quantity. Please see Appendix C.2 for a detailed proof. □

Proposition 2.1 indicates that the optimal capacity of MHMs and KHMs scales exponentially with $D_\Phi$. This capacity bound is provably tight and optimal for large feature dimension $D_\Phi$. It echos the exponential capacity lower bound in Lemma 2.1 and in prior works [Wu et al., 2024a,b, Hu et al., 2024a,b,c, 2023, Ramsauer et al., 2020]. Moreover, Lemma 2.2 shows that achieving the maximal capacity in any $D_\Phi$ is equivalent to achieving optimal codes. Thus, for a given memory set $\Xi$ of size $M$, the memory storage problem with KHMs divides into two sub-problems:

**(P1)** Finding a sufficiently large $D_\Phi$ (in Section 3.2), and

**(P2)** Finding a $\Phi$ such that $\Phi(\Xi)$ is an optimal spherical code (in Section 3.1).

Next, we examine these two sub-problems and present a sub-linear time algorithm to solve them.

## 3 Sub-Linear Time Algorithm for Optimal Memory Capacity

In this section, we present an sub-linear time algorithm that achieves optimal capacity. Then, we analyze the scaling behavior of $D_\Phi$ for KHMs to store any desired amount of memories.

### 3.1 Learning to Achieve Optimal Memory Code

Here we present an asymptotic result showing that an algorithm exists to find the optimal $\Phi$ for maximizing memory storage in dimension $D_\Phi$. Building on the results from the previous sections, we consider the following problem:

**Problem 1** (HardMax Problem). Given a memory set $\Xi = \{\xi_1, \ldots, \xi_M\}$, and assuming that $D_\Phi$ is sufficiently large to satisfy (2.2), we define the HardMax problem as finding a $\Phi$ such that $\Phi(\Xi)$ forms an optimal spherical code:

$$\min_{\Phi \in \mathcal{H}} \mathcal{L}_{\text{HardMax}}(\Phi), \quad \text{where} \quad \mathcal{L}_{\text{HardMax}}(\Phi) \coloneqq \max_{\nu, \mu \in [M], \nu \neq \mu} \langle \Phi(\xi_\mu), \Phi(\xi_\nu) \rangle \geq \rho^\star. \quad (3.1)$$

This problem setup involves finding a $\Phi$ such that $\Phi(\Xi)$ forms an optimal spherical code. Ideally, a more expressive function class $\mathcal{H}$ would simplify finding such a $\Phi$; exploring explicit forms of more powerful mappings is left for future work. Note that (3.1) represents a min-max optimization problem. Achieving the global optimum is notoriously challenging [Hsieh et al., 2021, Daskalakis et al., 2021, Shen et al., 2020]. Thus, we introduce a surrogate objective to solve (3.1):

**Definition 3.1** (Average Separation Loss). For $\tau > 0$, given a set of memory patterns $\Xi$ and a feature map $\Phi$, we define the average separation loss as

$$\mathcal{L}(\Xi, \tau, \Phi) \coloneqq \frac{1}{M} \sum_{\mu=1}^{M} \ell_\mu(\Xi, \Phi, \tau), \text{ where } \ell_\mu(\Xi, \Phi, \tau) \coloneqq \log \left[ \sum_{\nu=1}^{M} \exp \left( \frac{\langle \Phi(\xi_\mu), \Phi(\xi_\nu) \rangle}{\tau} \right) \right].$$

$$(3.2)$$

The primary difference between (3.1) and (3.2) is that (3.2) calculates average separation, whereas (3.1) focuses on the maximum separation between a single pair. This surrogate loss alleviates the challenging optimization, as (3.2) is convex. Therefore, with vanishing temperature $\tau$, the next theorem shows that (3.2) converges to the HardMax problem asymptotically.

**Theorem 3.1.** For any possible integer $M$, we have

$$\limsup_{\tau \to 0} \left( \underset{\Phi \in \mathcal{H}}{\arg\min} \, \mathcal{L}(\Xi, \Phi, \tau) \right) \subseteq \underset{\Phi \in \mathcal{H}}{\arg\min} \, \mathcal{L}_{\text{HardMax}}(\Phi).$$

**Algorithm 1** U-Hop+

---

**Input:** Iterations $N$, feature map $\Phi(x) \coloneqq Wx$, memory set $\Xi$, learning rate $\gamma \leq 1/G$ where $G$ is the Lipschitz constant of $\mathcal{L}$
**Output:** $x$

1: $W_0 \leftarrow W$
2: **for** $t = 0, ... N - 1$ **do**
3:     $W_{t+1} \leftarrow$ PGD $(W_t, \gamma, \Xi)$ .
4: **end for**
5: return $W_N$

---

*Proof.* we first introduce a helper function $\mathcal{L}_0$ in (C.10). We show that as $\tau \to 0$, $\mathcal{L}_0$ converges uniformly to $\mathcal{L}_{\text{HardMax}}$. Then, we prove that optimizing $\mathcal{L}_0$ and $\mathcal{L}$ yields the same optimal solution. Please see Appendix C.3 for a detailed proof. $\qquad\square$

Theorem 3.1 indicates that, with vanishing temperature, the minimiozation of (3.2) converges to the HardMax problem, i.e., their share the same optimal solution. This provide a theoretical justification for the empirical success of [Wu et al., 2024a]. In particular, the surrogate objective – the maximizing average separation between memories — leads to provably optimal memory capacity in low-temperature region (i.e., $\tau \to 0$). Lastly, we remark that that this analysis provides theoretical insights rather than practical guidance. To achieve high retrieval accuracy, the setting ($\tau = 1$) in [Wu et al., 2024a] is sufficient for a wide range of applications.

**U-Hop+: Sub-Linear Time Algorithm for Achieving Optimal Memory Capacity.** Next, we present Algorithm 1 for finding a $\Phi$ such that $\Phi(\Xi)$ forms an optimal spherical code. To meet the conditions in Definition 2.6, we use projected gradient descent to convert this constrained optimization problem into an unconstrained one. Several methods satisfy the requirements in Definition 2.6; we discuss them in Appendix B.2. We denote the learning rate as $\gamma$, the input matrix of the loss function as $X$, and the weight matrix as $W$. We define a single Projected Gradient Descent (PGD) step as

$$W_{t+1} = \text{PGD}(W_t, \gamma, X),$$

We defer the detailed formulation to Appendix B.2. Since the separation loss is convex and smooth, using projected gradient descent with a learning rate $\gamma \leq 1/G$, yields a sub-linear convergence rate of $\mathcal{O}(1/N)$ [Iusem, 2003]. This provides an asymptotic solution to the first sub-problem (P1). Next, we examine the relationship between feature dimension $D_\Phi$ and the number of memories $M$.

### 3.2 Impact of $D_\Phi$

This subsection analyzes the minimum $D_\Phi$ to store a given set of $M$ memories. Based on the *well-separation condition* and the derivation in Appendix C.2, the required $\Delta_{\min}^\Phi$ to store $M$ memories scales as $\mathcal{O}(\ln(M))$. With this insight, the following proposition shows the scaling behavior of required $D_\Phi$ with respect to $M$ and $\Delta_{\min}^\Phi$.

**Proposition 3.1.** Let $M^\star$ be the optimal memory capacity in $\mathbb{S}^{D_\Phi}$ and $\Phi \in \mathcal{H}$. For any optimal code $C^\star$ in $\mathbb{S}^{D_\Phi - 1}$ of size $M^\star$, the minimal separation $\rho(C^\star)$ is bounded by:

$$\frac{1}{2}\left(\frac{\sqrt{\pi}}{M^\star} \cdot \frac{\Gamma\left(\frac{D_\Phi+1}{2}\right)}{\Gamma\left(\frac{D_\Phi}{2}+1\right)}\right)^{\frac{2}{D_\Phi-1}} \leq \max_{\Phi \in \mathcal{H}} \Delta_{\min}^\Phi \leq 2\left(\frac{2\sqrt{\pi}}{M^\star} \cdot \frac{\Gamma\left(\frac{D_\Phi+1}{2}\right)}{\Gamma\left(\frac{D_\Phi}{2}\right)}\right)^{\frac{1}{D_\Phi-1}},$$

where $\Gamma(\cdot)$ is the gamma function.

**Remark 2.** By gamma function asymptotics, Proposition 3.1 is consistent with Proposition 2.1.

*Proof.* Please see Appendix C.4 for a detailed proof. $\qquad\square$

Proposition 3.1 establishes the separation value for memory codes that achieve optimal capacity in $D_\Phi$-dimensional space. Using this bound, for a given separation value $\Delta_{\min}^\Phi$, the minimum $D_\Phi$ required to store $M$ points scales as $\log\left(M^2/\Delta_{\min}^\Phi\right)$. We conduct an experiment to demonstrate the bound's tightness and provide an example with $D_\Phi = 3$ in Figure 3.

Table 1: **Distribution of Metastable State (in %).** For MNIST, we use the training set as memories and test set as queries. For synthetic data, we randomly generate the memories and queries. $\|p\|_0$ denotes the size of metastable state , which is the amount of non-zero entries of the probability distribution. For Softmax, we use a threshold of 0.01. For hyperparameter settings, see Table 3.

| | Synthetic | | | | | | MNIST | | | | | |
| | Softmax | | 1.5-entmax | | sparsemax | | Softmax | | 1.5-entmax | | sparsemax | |
| $\|p\|_0$ | - | U-Hop+ | - | U-Hop+ | - | U-Hop+ | - | U-Hop+ | - | U-Hop+ | - | U-Hop+ |
|---|---|---|---|---|---|---|---|---|---|---|---|---|
| 1 | 0.0 | 90.0 | 0.0 | 100.0 | 0.0 | 100.0 | 3.48 | 100.0 | 69.2 | 100.0 | 88.1 | 100.0 |
| 2 | 0.0 | 8.0 | 0.0 | 0.0 | 20.0 | 0.0 | 2.16 | 0.0 | 8.6 | 0.0 | 5.2 | 0.0 |
| 3 | 0.0 | 0.0 | 0.0 | 0.0 | 30.0 | 0.0 | 1.57 | 0.0 | 3.9 | 0.0 | 2.6 | 0.0 |
| 4 | 0.0 | 2.0 | 0.0 | 0.0 | 50.0 | 0.0 | 1.23 | 0.0 | 2.3 | 0.0 | 1.6 | 0.0 |
| 5 | 6.0 | 0.0 | 0.0 | 0.0 | 0.0 | 0.0 | 1.2 | 0.0 | 1.6 | 0.0 | 1.1 | 0.0 |
| 6 | 10.0 | 0.0 | 25.0 | 0.0 | 0.0 | 0.0 | 0.95 | 0.0 | 0.9 | 0.0 | 0.8 | 0.0 |
| 7 | 20.0 | 0.0 | 25.0 | 0.0 | 0.0 | 0.0 | 1.04 | 0.0 | 0.6 | 0.0 | 0.4 | 0.0 |
| 8 | 16.0 | 0.0 | 0.0 | 0.0 | 0.0 | 0.0 | 0.84 | 0.0 | 0.6 | 0.0 | 0.1 | 0.0 |
| 9 | 20.0 | 0.0 | 22.5 | 0.0 | 0.0 | 0.0 | 1.03 | 0.0 | 0.3 | 0.0 | 0.0 | 0.0 |
| $10^+$ | 28.0 | 0.0 | 27.5 | 0.0 | 0.0 | 0.0 | 86.5 | 0.0 | 12.0 | 0.0 | 0.1 | 0.0 |

## 4 Experimental Studies

### 4.1 U-Hop+ Reduces Metastable States

We compare the distribution of metastable state size under standard MHM and KHM update rules. The results are in Table 1. In general, with more metastable state having the size of 1, meaning the Hopfield model stores more memories as the query converges to a single memory. For metastable state size larger than 1, it represents that the retrieved pattern converges near the mean of a subset of memories, violating the requirement of $S_\mu^\Phi \cap S_\mu^\Phi = \emptyset$.

**Baselines.** We compare different variants of MHMs and KHMs. Santos et al. [2024], Wu et al. [2024b] provide comprehensive analyses of modern Hopfield models with various normalization functions. Here, we consider softmax, 1.5-entmax, and sparsemax for normalization. We equip these three baselines with U-Hop to compare against standard MHMs.

**Settings and Metrics.** Let $p = \text{Softmax}(\beta x^\mathsf{T} \Xi)$. We determine whether the update rule converges to either a single memory or a mixture of memories by observing the probability distribution $p$. The quantity $\|p\|_0$ represents the size of the metastable state, which is the number of non-zero entries in the probability distribution. In the case of 1.5-entmax and sparsemax, we calculate $\|p\|_0$ directly. For softmax, since it only generates non-zero entries, we use a threshold of $0.01$ and consider the entries under the threshold as 0. We conduct experiments using both synthetic and MNIST datasets. For MNIST, we use the training set as memories and the test set as queries. For synthetic datasets, we randomly generate memories and queries with Gaussian initialization. To ensure the convergence to the fixed point, we perform multiple updates on the query. For more details, refer to Appendix D.1.

**Results.** On both synthetic and MNIST datasets, it is evident that under separation maximization, the size of the metastable state dramatically decreases within just 20 iterations of Algorithm 1. This result demonstrates that, with Algorithm 1, KHMs are capable of storing patterns that MHMs cannot store. The significant percentage of size 1 metastable states in KHMs indicates that they circumvent the memory confusion problem in dense associative memory models [Krotov and Hopfield, 2016]. For the MNIST dataset, we see MHMs show close performance with KHMs under 1.5-entmax and sparsemax, showing that the methods in [Santos et al., 2024, Wu et al., 2024b, Hu et al., 2023] also circumvent the memory confusion problem. Notably, KHMs require only one-fourth of the dimensions to store memories while perfectly storing 60,000 MNIST patterns. These results suggest that KHMs with Algorithm 1 efficiently utilize feature dimensions for memory storage.

### 4.2 Energy Landscape under U-Hop+ Stores More Memories

**Settings and Metrics.** We visualize the energy landscape of KHMs at different stages of Algorithm 1 using contour plots. The results are presented in Figure 1. We consider two settings: 2 and 4 memories stored in a 2-dimensional space. Ideally, the energy landscape should position memories in multiple separated low-energy regions (valley), with each region isolated from others by high-energy regions. If multiple memories share the same valley, it leads to memory confusion and the presence of metastable states during energy minimization. For experiment details, refer to Appendix D.2.

**Results.** The first row in Figure 1 shows the raw energy landscape without KHM and Algorithm 1, corresponding to the modern Hopfield energy landscape. On the right side of Figure 1, we observe that MHMs are only able to store 2 out of 4 points, but KHMs are able to further separate one point

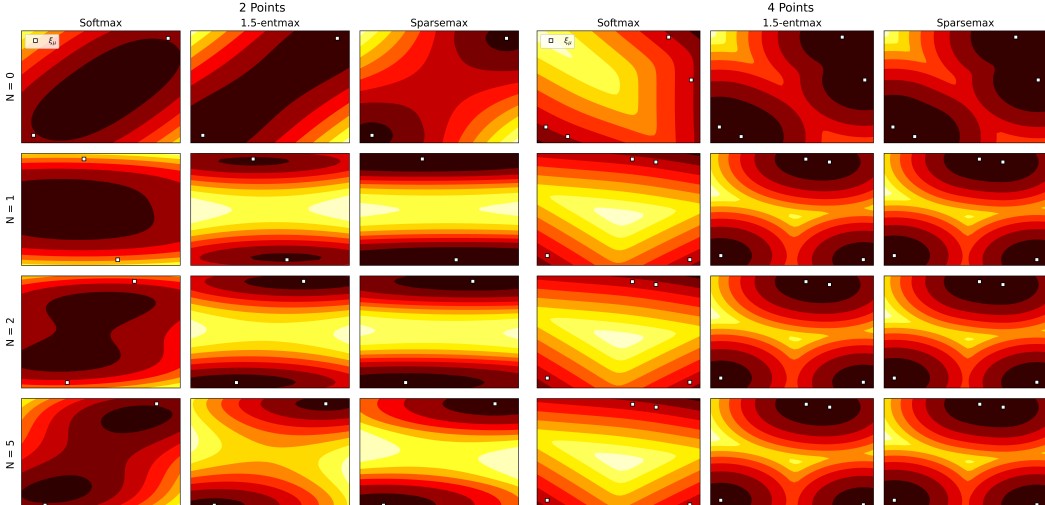

Figure 1: **Energy Landscape under Different Iterations of Algorithm 1.** Left: $M = 2$, Right: $M = 4$. Lighter color represents higher energy. The first row represents the raw energy landscape without applying U-Hop+. The second to last row represents the energy landscape when $N = (1, 2, 5)$. The visualization shows that Algorithm 1 not only separates the local minima better, but also pushes memories closer to the fixed point.

Table 2: **Test AUC of Multiple Instance Learning Datasets.** We compare the HopfieldPooling-based model with and without U-Hop+. We use the dense [Ramsauer et al., 2020] and sparse [Hu et al., 2023] modern Hopfield models as baselines. We use $K$-fold cross validation on all 4 datasets, with $K = 10$. The reported AUC is the average AUC score across 10 folds. For the baselines, we use the results reported in [Hu et al., 2023]. For our method, we directly use the default hyperparameter without grid search instead of using hyperparameter optimization (HPO) in [Hu et al., 2023, Ramsauer et al., 2020]. We exclude the variance as they are all smaller than $0.07$. The result shows that even without HPO, U-Hop+ is still able to obtain a performance gain.

| Method | Tiger | Elephant | Fox | UCSB |
|---|---|---|---|---|
| Modern Hopfield | 0.871 | 0.876 | 0.637 | 0.828 |
| Modern Hopfield + U-Hop+ | 0.881 | 0.921 | 0.648 | 0.831 |
| Sparse Hopfield | 0.884 | 0.914 | 0.610 | 0.796 |
| Sparse Hopfield + U-Hop+ | 0.887 | 0.921 | 0.638 | 0.805 |

from the others, resulting in memorizing one extra pattern. Lemma 2.2 and Theorem 3.1 indicate that U-Hop+ pushes memories away from each other, providing more isolated $S_\mu^\Phi$, for $\mu \in [M]$. We observe this phenomenon across all settings, especially under the 2-point configuration with Softmax and $1.5$-entmax, where the low-energy region is split into two distinct valleys as $N$ increases. This process shows how U-Hop$^+$ is able to store memories that MHMs cannot. With energy minimization, the query converges to either one of the minima instead of the mixture thereof (also showed in Figure 2). Additionally, we also notice that the contour lines exhibit steep slopes between different local minima in the 2-point setting under $1.5$-entmax and sparsemax. This implies that U-Hop+ pushes local minima further away from each other and deepens each one of them. Such sharp changes in energies lead to faster convergence to fixed points due to larger gradients.

**Basins of Attraction.** Figure 2 shows the basins of attraction of queries w.r.t. MHM and KHM under the scenario of storing 5 patterns. We randomly initialize 5 patterns with normal distribution. We run the update rule for 5 iterations and see whether each query converges to a single memory (colored) or to a metastable state (white). Following the above setting, we track the attraction basins throughout each iteration of Algorithm 1. We defer more details to Appendix D.3. Specifically, most MHM variants are not capable of converging to fixed points in 5 updates. While U-Hop+ dramatically improves such aspect, where most queries are able to converge either one of the memories. Moreover, the increased $R_\Phi$ also leads to a larger $S_\mu^\Phi$, making more queries to converge to a single memory. Additionally, there is a performance gap between Softmax ($\alpha = 1$) and other sparse variants, which matches the findings in [Santos et al., 2024, Hu et al., 2023].

### 4.3 Multiple Instance Learning

We conduct multiple instance learning (MIL) on 4 real-world datasets using Hopfied-based models with and without our U-Hop+ algorithm. We follow the setup in [Santos et al., 2024, Wu et al., 2024a, Hu et al., 2023] by using a model with 1 embedding layer, 1 HopfieldPooling layer and a linear readout layer. We first utilize Algorithm 1 to "pretrain" the embedding and HopfieldPooling layer,

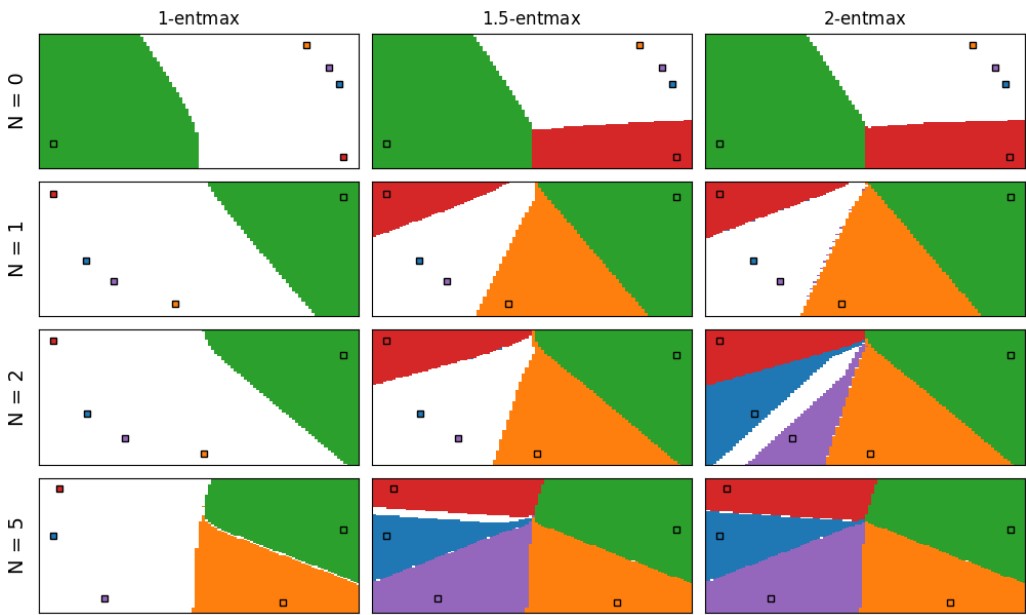

Figure 2: **Basins of Attraction Comparison of** Algorithm 1. The first row represents the raw Basins of Attraction without applying U-Hop+ or KHM. The second to last row shows the basins when $N = (1, 2, 5)$. Square points are memories. White area is where queries are not able to converge to a single memory. Colored area is where queries converges to the corresponding memory. The result indicates that U-Hop+ is capable of converging to fixed point fast and reduce metastable states. 1 and 2-entmax corresponds to Softmax [Ramsauer et al., 2020] and Sparsemax [Hu et al., 2023].

and then fine-tune the whole model on the MIL task. The results are in Table 2. We observe that both dense and sparse Hopfield-based models obtain performance boost when equipped with U-Hop+, indicating our method is also effective in practical scenarios. Further, as demonstrated in Figure 5, the separation loss converges fast, indicating U-Hop+ is a lightweight method for performance boost.

## 5   Discussion and Conclusion

This work complements U-Hop [Wu et al., 2024a] by establishing the optimal capacity of kernelized modern Hopfield models (KHMs) and providing the first tight, optimal memory capacity bound for transformer-compatible dense associative memories. We start by connecting stored memories in KHMs to spherical codes from information theory. We then prove that maximizing memory storage in KHMs requires arranging memories as an optimal spherical code in feature space. This allows us to matches the well-known exponential lower bound [Wu et al., 2024a,b, Hu et al., 2024a,b,c, 2023, Ramsauer et al., 2020] with an upper bound. This achievement is notable, as deriving such a tight bound is challenging due to the max-min structure of maximal separation among stored memories [Wu et al., 2024a, Section 5]. Moreover, we introduce a sub-linear time algorithm to achieve this optimal capacity, U-Hop+ (Algorithm 1). U-Hop+ performs this rearrangement with a convergence rate of $\mathcal{O}(1/N)$. Additionally, we analyze the minimum dimension $D_\Phi$ required to store $M$ memories. Numerically, we validate the effectiveness of KHMs and demonstrate how Algorithm 1 enhances memory storage in both KHM retrieval tasks and transformer representation learning tasks.

**Can U-Hop+ Preserve Semantic Meanings?** In representation learning, it is crucial to preserve relationships in the feature space after encoding data [Wang et al., 2023, Neelakantan et al., 2022]. The primary strategy is to ensure the embeddings of similar instances share similar directions in Euclidean space. At first glance, the approach of pushing all memories away from each other in Equation (3.2) may seem counterintuitive. However, as detailed in Appendix D.5, we find that the learned feature map still encodes similar instances closely together (Figure 4), even without semantic information involved. This result indicates that U-Hop+ stores memories in a semantically coherent manner. A discussion of the separation capability of $\Phi$ can be found in Figure 4.

**Limitations.** One limitation of our work only considers linear affine functions as the feature map $\Phi \in \mathcal{H}$. Additionally, standard spherical code analysis focuses only on normalized points on a hypersphere, ignoring memories with varying magnitudes. We leave them for future research.

## Broader Impact

We expect no negative social impacts as this work mostly present theoretical results and numerical simulations. As discussed in our introduction, this paper develops a theoretical framework to study Kernelized Hopfield models, potentially benefit the area of computational associative (Hopfield) memory models, transformer networks and large foundation models.

## Acknowledgements

JH thanks Thomas Burns, Dmitry Krotov, Mimi Gallagher, Sara Sanchez, Dino Feng, and Andrew Chen for enlightening discussions; Robin Luo, Jiahao Yu, Weimin Wu, and Teng-Yun Hsiao for collaboration on related topics; the Red Maple Family for their support; and Jiayi Wang for facilitating experimental deployments. The authors also thank the anonymous reviewers and program chairs for their constructive comments.

JH is partially supported by the Walter P. Murphy Fellowship. DW is supported by NIH R01LM1372201. HL is partially supported by NIH R01LM1372201, AbbVie and Dolby. This research was supported in part through the computational resources and staff contributions provided for the Quest high performance computing facility at Northwestern University which is jointly supported by the Office of the Provost, the Office for Research, and Northwestern University Information Technology. The content is solely the responsibility of the authors and does not necessarily represent the official views of the funding agencies.

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

# Appendix

# A   Related Works

**Hopfield and Dense Associative Memory Models.**   Associative memory models [Kanerva, 1988, Willshaw et al., 1969] are extensively studied in neuroscience and machine learning due to their biologically plausible designs. These models aim to store a set of memories and accurately retrieve each one given an input query.  Hopfield models [Hopfield, 1982] are a class of energy-based associative memory models, beginning with classical versions [Hopfield, 1984, 1982] that handle binary patterns and have a (sub-)linear memory capacity of $\mathcal{O}(d)$ for a pattern dimension $d$. Dense associative memory models [Krotov and Hopfield, 2021, 2016, Ramsauer et al., 2020, Demircigil et al., 2017] are later proposed with superlinear memory capacity enabled by sharper energy functions (e.g., polynomial and exponential). Notably, the latest advancement in these dense models is their exponential-in-$d$ capacity [Lucibello and Mézard, 2024, Santos et al., 2024, Wu et al., 2024b, Hu et al., 2024a,c,b, 2023, Ramsauer et al., 2020]. However, the current literature reports are mostly[6] lower bound results for this exponential memory capacity. This work matches these lower bounds with an upper bound, making the capacity both tight and provably optimal.

**Transformer-Compatible Dense Associative Memories: Modern Hopfield Models.**   Recently, a special class of dense associative memory models, modern Hopfield models (MHMs), has gained increasing interest in deep learning due to their connection to the attention mechanism in transformers [Wu et al., 2024a,b, Hu et al., 2024a,c,b, 2023, Ramsauer et al., 2020]. Therefore, we also refer to them as *transformer-compatible* dense associative memories. Notably, the defining characteristic of MHMs is that their single-step update is equivalent to the attention mechanism [Vaswani et al., 2017]. This striking feature makes them interesting given the prevalence and dominance of transformer architectures in the era of large foundation models [Polyak et al., 2024, OpenAI, 2024, Hu et al., 2024d, Peebles and Xie, 2023, Bubeck et al., 2023, Bai et al., 2023, Achiam et al., 2023, Zhou et al., 2024, 2023, Ji et al., 2021, Touvron et al., 2023, Kojima et al., 2022, Bommasani et al., 2021, Radford et al., 2019, Devlin, 2018].

As a result, such a connection facilitates the integration of associative memory models into modern deep learning and large foundation models [Burns, 2024, Xu et al., 2024, Hu et al., 2024a,b, Hofmann et al., 2024, Wu et al., 2024b, Auer et al., 2023, Burns and Fukai, 2023, Fürst et al., 2022, Krotov and Hopfield, 2016]. Moreover, recent studies introduce several modern Hopfield model variants. Hu et al. [2023] propose the sparse modern Hopfield model, a sparse counterpart to MHM with larger capacity and lower retrieval error. Wu et al. [2024b], Santos et al. [2024], Martins et al. [2023] introduce generalized sparse Hopfield models that unify all MHMs with different degrees of sparsity. Hu et al. [2024b] present a nonparametric construction for deep learning compatible Hopfield layers and several efficient modern Hopfield variants. Hu et al. [2024a] introduce OutEffHop, an outlier-removing, deep learning compatible Hopfield layer for robust large pretrained model quantization. Wu et al. [2024a] propose U-Hop facilitating memory retrieval in a learnable feature space (see introduce for a review). Empirically, U-Hop improves the memory confusion problem [Krotov and Hopfield, 2021] by a significant margin.

**Applications of Modern Hopfield Models in Modern Machine Learning.**   Recently, modern Hopfield models also achieve empirical success across various deep learning tasks [Krotov, 2023]. Starting with [Krotov and Hopfield, 2016], the polynomial Hopfield model is proposed for image classification tasks (e.g., MNIST). Later, Ramsauer et al. [2020] introduce modern Hopfield layers compatible with deep learning architectures. Since then, various modern Hopfield layers [Hu et al., 2024a, Wu et al., 2024b, Hu et al., 2023] have been applied in many deep learning tasks, such as language modeling [He et al., 2024], multiple instance learning [Ramsauer et al., 2020], immune repertoire classification [Widrich et al., 2020], multivariate time series prediction [Wu et al., 2024b], image generation [Hoover et al., 2023], tabular learning [Xu et al., 2024], unsupervised clustering [Saha et al., 2023], and image captioning [Fürst et al., 2022].

Additionally, modern Hopfield layers introduce new operations in large foundation models to improve performance. For example, Wu et al. [2024b] leverage the retrieval dynamics of modern Hopfield models to enable an external memory plugin in time series prediction, and Xu et al. [2024] apply a

---

[6]After completion of this work, the authors attended ICML 2024 and learned of an upper bound result reported by [Santos et al., 2024], which also utilizes techniques from spherical codes. The authors regret missing the dinner with Santos et al. [2024] at Vienna due to a tight schedule. The difference between this work and [Santos et al., 2024] lies not only in scope but also in that our upper bound matches the lower bound in the low-temperature region (Proposition 2.1).

similar approach to tabular data. In image captioning, Fürst et al. [2022] address issues with covariance structure using modern Hopfield models. Hu et al. [2024a] propose an outlier-free Hopfield layer as a quantization-strong and resource-efficient transformer backbone for large language models and large foundation models. Ota et al. [2023] embed a partial forgetting functionality in modern Hopfield models to enhance model performance.

**Kernelized Hopfield Models.** Wu et al. [2024a] propose kernelized Hopfield models (KHMs)[7], which have the capability to store memories in a learnable feature space. KHMs offer the flexibility to relocate memories while maintaining several defining properties of modern Hopfield models [Ramsauer et al., 2020, Theorems 1 to 4]. By maximizing the average separation between memories, KHMs empirically achieve lower retrieval errors [Wu et al., 2024a, Section 4]. However, the theoretical understanding of KHMs is larking due to their new flexibility in rearranging memories. This work aims to fill this gap by analyzing the capacity limits and theoretical justifications for KHMs.

**Spherical Code.** Spherical codes are mathematical constructions describing the arrangement of $M$ points on the surface of a $d$-dimensional hyper-sphere [Delsarte et al., 1991]. The main problem around spherical codes is to arrange points in a way such that the minimum distance between any two points are maximized. This arrangement is called the optimal spherical code. It is crucial for minimizing errors and maximizing efficiency in signal transmission and data storage. In general, the value of max minimal separation and arrangement of points is unsolved except for certain pairs of $(d, M)$ [Wang, 2009]. Various of fields such as communications [Strohmer and Heath Jr, 2003] and data compression [Lelewer and Hirschberg, 1987]. Moreover, spherical codes are related to the area of error-correcting codes [Peterson and Weldon, 1972], which are used to detect and correct errors in data transmission and storage.

# B  More Discussions

## B.1  MHM Capacity

We review the modern Hopfield capacity lower bound in [Ramsauer et al., 2020].

**Lemma B.1** (Memory Capacity of MHM). Let $1 - p$ be the probability of successfully storing and retrieving a pattern. Assuming patterns are normalized, the amount of patterns randomly sampled from a $d$-dimensional unit-sphere that the MHM with update rule in (1.2), can store and retrieve is lower-bounded by

$$M \geq \sqrt{p}C^{(d-1)/4},$$

where $C$ is the solution to $C = {}^{b}/W_0(\exp\{a + \ln b\})$, with $W_0(\cdot)$ being the principal branch of Lambert $W$ function, $a := ({}^{4}/d-1)\left(\ln {}^{2\sqrt{p}-2}/R + 1\right)$ and $b := {}^{4\beta}/5(d-1)$.

Observe $W_0$ is an increasing function in $\exp\{a + \ln b\}$, indicating that $C$ is increasing in $R$ ( $C$ is increasing in $a$; $a$ is decreasing with fixed $R$). Finally, we observe that under Assumption 1, we have $R = \frac{1}{2}\sqrt{2\Delta_{\min}}$, implying that the memory capacity of MHM is constrained by large $\Delta_{\min}$.

## B.2  Learning on Stiefel Manifolds

We review ways to do optimization on the surface of an unit-hypersphere (Stiefel manifold).

**Projected Gradient Descent.** Given any loss function $\mathcal{L}(\cdot)$, an input matrix $X$, and learning rate $\gamma$, a single gradient descent step at $t$-th time step is:

$$W_{t+1/2} = W_t - \gamma_t \nabla_X \mathcal{L}(X). \tag{B.1}$$

Projected gradient descent [Raman and Yang, 2019] then projects $W_{t+\frac{1}{2}}$ onto the feasible set, in this case, the surface of a unit hypersphere

$$W_{t+1} = \frac{W_{t+1/2}}{\left\|W_{t+1/2}\right\|}. \tag{B.2}$$

---

[7]This is different from [Iatropoulos et al., 2022] while they share similar names. As an independent yet related effort, Hoover et al. [2024a] propose the DrDAM (Distributed Representation for Dense Associative Memory) as a kernel approximation of the standard dense associative memory models based on random feature decomposition.

Combining (B.1) and (B.2), we obtain the projected gradient descent step as

$$W_{t+1} = \texttt{PGD}(W_t, \gamma, X).$$

**Riemannian Optimization.** [Tripuraneni et al., 2018] construct and analyze an approach on optimization on Riemannian manifolds from a geometric perspective. Their adapt the *Polyak-Ruppert* [Polyak and Juditsky, 1992] iterate averaging technique to the Riemannian setting. In general, with carefully selected step-size, their method achieves $\mathcal{O}(1/N)$ convergence rate which is the same in the euclidean setting. Overall, there are various of methods for optimization on Riemannian manifolds with comparable convergence rate to Euclidean space. Thus, giving the advancement of Riemannian optimization, our assumption in Definition 2.6 is reasonably mild.

**L2 Regularization.** A simple alternative to satisfy the norm constraint is through L2 regularization. From the aspect of neural collapse [Papyan et al., 2020], learning under (3.2) is similar to learning under cross-entropy loss or supervised contrastive learning without positive samples [Graf et al., 2021]. Further, from the analysis on unconstrainted feature models [Zhu et al., 2021], we can see that with carefully chosen coefficient on the regularization term on $W$, the optimal solution of (3.2) or cross-entropy loss ended up outputting normalized features.

Based on the previous research, we are able to see that the constraint in Definition 2.6 is not difficult to satisfy.

### B.3 KHMs as Brain-Machine Interface

The ultimate goal in the field of Brain-Machine Interface (BMI) [Lebedev and Nicolelis, 2006] is to design communication between human brains and external devices. It has potential in various real-world applications such as medical treatment [Ramos-Murguialday et al., 2013, Shanechi, 2019], virtual reality [Lécuyer et al., 2008], etc. While Hopfield models serve as computational models for simulating human brains and their memory recall system [Liu et al., 2015, Hsu, 2012], KHMs correspond to external devices (storage space) to assist/enhance the process of memory recall.

In this paper, we study the scenario where we try to optimize the usage of external neurons for memory storage by increasing the separation between memories in such external space ($\Phi$-space). We show that to prevent memory confusion, minimizing separation loss is an effective way to utilize the external space efficiently. The closest work we can find is [Morrison et al., 2017] and [Kozachkov et al., 2023]. In particular, they use Hopfield (and modern Hopfield or dense associative memory) models as the computational model for human brain. Specifically, Morrison et al. [2017] study the case of memory loss caused by disease or injuries. Their proposed framework is able to consider the level of memory damage and then estimate the required dimension to fully or partially recover memories. Kozachkov et al. [2023] explore the potential of building biological computers through links between transformers and modern Hopfield models. They show that neuron–astrocyte networks can perform the core computations of a transformer. In this context, our work provide an improved connection between brain and transformer models, offering a more learnable and powerful approach to brain-machine interfaces (BMI).

# C  Proofs of Main Text

## C.1  Proof of Lemma 2.1

We first introduce a helper lemma:

**Lemma C.1** ([Ramsauer et al., 2020, Hu et al., 2023]). Given real numbers, $a, b \in \mathbb{R}$. If the equation

$$ac + c \ln c - b = 0,$$

holds, then the solution is

$$c = \frac{b}{W_0(\exp\{a + \ln b\})}.$$

By [Hu et al., 2023, Corollary 3.1.1], we state the well-separation condition of dense modern Hopfield model [Ramsauer et al., 2020].

**Lemma C.2** (Well Separation Condition of Dense Modern Hopfield Model [Ramsauer et al., 2020]). Following Definition 2.2, suppose the memory patterns $\{\xi_\mu\}_{\mu \in [M]}$ are located within the sphere $S_\mu := \{x \mid \|x - \xi_\mu\| \leq R\}$. Then, assuming normalized memory patterns, the retrieval dynamics $\mathcal{T}_{\text{MHM}}$ maps the sphere $S_\mu$ onto itself under the following conditions:

1. The initial query $x$ is located within the sphere $S_\mu$, i.e. $x \in S_\mu$.
2. The *well-separation* condition is satisfied, which is given by:

$$\Delta_\mu \geq \frac{1}{\beta} \ln \frac{2(M-1)}{R} + 2R.$$

This specifies the necessary condition for a pattern $\xi_\mu$ to be stored in $E$ and be able to retrieved by $\mathcal{T}$.

*Proof of Lemma 2.1.* Let $\Delta_{\min}^\Phi = \text{Min}_{\mu \in [M]} \Delta_\mu^\Phi$, and $\theta_{\mu\nu,\Phi}$ be the angle between two patterns $\Phi(\xi_\nu)$ and $\Phi(\xi_\mu)$. Note that $\theta_{\Phi,\mu\nu} \in [0, \pi]$.

By Definition 2.7, we have

$$\Delta_{\min}^\Phi \geq \frac{1}{\beta} \ln \left( \frac{2(M-1)}{R_\Phi} \right) + 2R_\Phi,$$

and

$$\Delta_{\min}^\Phi = \text{Min}_{1 \leq \mu \leq \nu \leq M} \left( 1 - \cos(\theta_{\Phi,\mu\nu}) \right) = \left[ 1 - \cos(\theta_{\min}) \right],$$

where $\theta_{\min} := \text{Min}_{1 \leq \mu \leq \nu \leq M} \left( 1 - \cos(\theta_{\Phi,\mu\nu}) \right) \in [0, \pi]$.

Then, it holds

$$[1 - \cos(\theta_{\min})] \geq \frac{1}{\beta} \ln \left( \frac{2(M-1)}{R_\Phi} \right) + 2R_\Phi. \tag{C.1}$$

Next, Let $1 - p$ be the success storage and retrieval probability under Definition 2.7. We have

$$P \left( \Delta_\mu^\Phi \geq \frac{1}{\beta} \ln \left( \frac{2(M-1)}{R_\Phi} \right) + 2R_\Phi \right) = 1 - p.$$

By (C.1), we have

$$P \left( 1 - \cos(\theta_{\min}) \geq \frac{1}{\beta} \ln \left( \frac{2(M-1)}{R_{\Phi,\mu}} \right) + 2R_\Phi \right) = 1 - p. \tag{C.2}$$

We observe that $\cos(\theta_{\min})$ connects to the maximal separation loss via

$$\cos(\theta_{\min}) = \frac{\mathcal{L}_\Phi(\Xi) + 2t}{2t}.$$

Following the proof of [Hu et al., 2023, Lemma 3.1] and by Lemma C.1, it holds

$$M = \sqrt{p}C^{d-1/4},$$

with some real value $C \in \mathbb{R}$. Here $C$ is solution to the upper branch of the Lambert $W$ function deduced from (C.2),

$$C = \frac{b}{W_0(\exp\{a + \ln b\})}, \tag{C.3}$$

where

$$a := \frac{4}{d-1}\left\{\ln\left[\frac{2(\sqrt{p}-1)}{R_\Phi}\right] + 1\right\}, \quad \text{and} \quad b := \frac{4\beta}{5(d-1)}. \tag{C.4}$$

Then, we arrive a lower bound on the exponential storage capacity M:

$$M \geq \sqrt{p}C^{\frac{d-1}{4}}. \tag{C.5}$$

To compare the the results from [Ramsauer et al., 2020, Theorem 3] (with the assumption of pattern normalization), we denote the results from [Ramsauer et al., 2020] with $\widetilde{\cdot}$ notation, i.e.

$$\widetilde{a} := \frac{4}{d-1}\left\{\ln\left[\frac{2(\sqrt{p}-1)}{R}\right] + 1\right\}, \quad \text{and} \quad \widetilde{b} = b.$$

And we also have $\widetilde{\theta}_{\min} := \text{Min}_{1 \leq \mu \leq \nu \leq M}(1 - \cos(\theta_{\mu\nu})) \in [0, \pi]$ be the angle between two raw memory patterns $\xi_\nu, \xi_\mu$.

We denote the optimal separation loss be $\mathcal{L}^\star(\Xi)$, and the loss value at $t$-th step be $\mathcal{L}^t(\Xi)$.

We denote $R_\Phi^\star$ be the corresponding $R_\Phi$ when $\mathcal{L}_\Phi(\Xi)$ is at its global minimum.

By (3.2), the convexity of $\mathcal{L}_\Phi$, the optimality of $\mathcal{L}_\Phi$ gives

$$R_\Phi^\star = \frac{1}{2}\sqrt{\frac{\mathcal{L}_\Phi^\star(\Xi)}{-t}} \geq R_\Phi.$$

Next, we prove that to achieve $R_\Phi \geq R$, we need $\mathcal{O}\left(\frac{1}{-4tR^2 - \mathcal{L}_\Phi^\star(\Xi)}\right)$ sub linear time (iterations.).

Recall that $R := \frac{1}{2}\text{Min}_{\nu,\nu\neq\mu}\|\xi_\nu - \xi_\mu\|$. By $R_\Phi = \sqrt{\mathcal{L}_\Phi(\Xi)/-t}/2$, for $R_\Phi \geq R$, we need

$$\frac{1}{2}\sqrt{\frac{\mathcal{L}_\Phi(\Xi)}{-t}} \geq \frac{1}{2}\text{Min}_{\nu,\nu\neq\mu}\|\xi_\nu - \xi_\mu\|,$$

which implies, by $t > 0$,

$$\mathcal{L}_\Phi(\Xi) \leq -t \cdot (2R)^2. \tag{C.6}$$

Subtracting $-\mathcal{L}_\Phi^\star(\Xi)$ on both sides, we get

$$\mathcal{L}_\Phi(\Xi) - \mathcal{L}_\Phi^\star(\Xi) \leq -t \cdot (2R)^2 - \mathcal{L}_\Phi^\star(\Xi) := \epsilon$$

Which implies the iteration number needed to achieve improved memory capacity bound is:

$$N = \mathcal{O}\left(\frac{1}{-4tR^2 - \mathcal{L}_\Phi^\star(\Xi)}\right),$$

which gives us a sub-linear time complexity.

Let

$$a := \frac{4}{d-1}\left\{\ln\left[\frac{2(\sqrt{p}-1)}{R_\Phi}\right] + 1\right\}, \quad \text{and} \quad b := \frac{4\beta}{5(d-1)}.$$

As long as Algorithm 1 runs for $\mathcal{O}\left(1/-4tR^2 - \mathcal{L}_\Phi^\star(\Xi)\right)$ iterations, its output $\Phi$ satisfies

$$\widetilde{a} \le a, \tag{C.7}$$

and

$$\widetilde{C} = W_0\left(\exp\left\{\widetilde{a} + \ln\widetilde{b}\right\}\right) \le W_0\left(\exp\{a + \ln b\}\right) = C. \tag{C.8}$$

Thus, we have the memory capacity comparison as

$$M = \sqrt{p}C^{\frac{d-1}{4}} \ge \sqrt{p}\widetilde{C}^{\frac{d-1}{4}} = \widetilde{M}.$$

Since the upper branch of Lambert W function is monotonically increasing on its domain, $R_\Phi > R$ implies

$$M = \sqrt{p}C^{\frac{d-1}{4}} \ge \sqrt{p}\widetilde{C}^{\frac{d-1}{4}} = \widetilde{M}.$$

Hence we finish the proof. $\qquad\square$

## C.2 Proofs of Lemma 2.2 and Proposition 2.1

**Lemma C.3** (Capacity of Optimal Spherical Code, Lemma 2.2 Restated). Given a fixed $D_\Phi > 1$, and its corresponding $M^\star$, if an optimal code $\mathcal{C}_{\text{opt}}$ is in $\mathbb{S}^{D_\Phi - 1}$ and has size $M^\star$, then $\mathcal{C}_{\text{opt}} \in \Lambda_{D_\Phi}$.

**Proposition C.1** (Optimal Memory Capacity, Proposition 2.1 Restated). Following Lemma 2.2, we have

$$M^\star \asymp c^{D_\Phi},$$

for some $c > 1$. Here $\asymp$ indicates matching upper and lower bounds up to constant factors.

*Proof.* By Assumption 1, all memories are normalized. Thus, we have

$$R_\Phi = \frac{1}{2}\sqrt{2 - 2\max_{\substack{\mu,\nu \in [M] \\ \mu \neq \nu}} \langle \Phi(\xi_\mu), \Phi(\xi_\nu) \rangle} \hspace{2cm} \text{(By (2.1))}$$

$$= \sqrt{\frac{1}{2}\Delta_{\min}^\Phi}. \hspace{2cm} \text{(C.9)}$$

Recall the storage condition

$$\Delta_\mu^\Phi \geq \frac{1}{\beta}\ln\left(\frac{2(M-1)}{R_\Phi}\right).$$

Here we consider the minimal $\Delta_\mu^\Phi$ among all possible $\mu \in [M]$. We plug (C.9) into the well-separation condition and change $\Delta_\mu^\Phi$ to $\Delta_{\min}^\Phi$. We arrive

$$\Delta_{\min}^\Phi \geq \frac{1}{\beta}\ln\left(\frac{2(M-1)}{\sqrt{\Delta_{\min}^\Phi/2}}\right).$$

By rearranging terms, we get

$$\Delta_{\min}^\Phi + \frac{1}{2\beta}\ln\left(\frac{1}{2}\Delta_{\min}^\Phi\right) \geq \frac{1}{\beta}\ln\left(2(M-1)\right).$$

The derivative w.r.t. $\Delta_{\min}^\Phi$ on the LHS is

$$1 + \frac{1}{2\beta\Delta_{\min}^\Phi},$$

indicating that LHS is increasing in $\Delta_{\min}^\Phi$ for all $\Delta_{\min}^\Phi > 0$. The derivative w.r.t. $M$ on the RHS is

$$\frac{1}{\beta(M-1)},$$

indicating that the RHS is increasing in $M$ for all $M > 1$. Since we are handling $\Delta_{\min}^\Phi$, this property holds for all $\Delta_\mu^\Phi$.

Let $\delta$ be the minimum value for $\Delta_{\min}^\Phi$ that satisfies the storage condition such that

$$\delta + \frac{1}{2\beta}\ln\left(\frac{1}{2}\delta\right) \geq \frac{1}{\beta}\ln\left(2(M-1)\right).$$

With the definition of optimal spherical code, we have $\delta \leq 1 - \rho^\star$. Thus an optimal spherical code must satisfy this inequality.

Now we further analyze the quantity $M^\star$. Let $\theta = \arccos\left(\rho(C_{\text{opt}})\right)$, with $\theta \in (0, \pi/2)$, we apply the upper bound in [Kabatiansky and Levenshtein, 1978], we get

$$e^{\varphi(\theta)D_\Phi(1+o(1))} \geq M^\star.$$

where $\Phi \in \mathcal{H}$. With the above result, we get $M^\star = o(c^{D_\Phi})$ for some $c > 1$.

For the lower bound of $M^\star$, we use the classic sphere code bound in [Chabauty, 1953, Shannon, 1959, Wyner, 1965], and get

$$M^\star \geq \left[\frac{1}{\sqrt{\pi}} \frac{\Gamma(D_\Phi/2)}{\Gamma(D_\Phi-1/2)} \int_0^\theta \sin^{D_\Phi-2}(x)dx\right]^{-1} = (1+o(1))\sqrt{2\pi D_\Phi} \cdot \frac{\cos(\theta)}{\sin^{D_\Phi-1}(\theta)},$$

where $x \in \mathbb{S}^{D_\Phi-1}$.

Therefore, we have

$$e^{\varphi(\theta)D_\Phi(1+o(1))} \geq M^\star \geq (1+o(1))\sqrt{2\pi D_\Phi} \cdot \frac{\cos(\theta)}{\sin^{D_\Phi-1}(\theta)},$$

where $\varphi(\theta) > -\log\sin(\theta)$, and $o(\cdot)$ is "strictly slower than" notation as $D_\Phi \to \infty$.

This completes the proof. Tighter bounds can be found in [Jenssen et al., 2018, Fernández et al., 2021]. We selected bounds that most clearly show the exponential scaling behavior for better intuition. □

## C.3 Proof of Theorem 3.1

We first restate Theorem 3.1:

**Theorem C.1** (Theorem 3.1 Restated). For any possible integer $M$, we have

$$\limsup_{\tau \to 0} \left( \operatorname*{argmin}_{\Phi \in \mathcal{H}} \frac{1}{M} \sum_{\mu=1}^{M} \mathcal{L}_\Phi(\xi_\mu, \tau) \right) \subseteq \operatorname*{argmin}_{\Phi \in \mathcal{H}} \mathcal{L}_{\text{HardMax}}(\Phi),$$

where all $\mathcal{H}$ is the hypothesis space of $\Phi$.

Then we introduce a helper lemma.

**Lemma C.4.** Let $\mathcal{L}_0(\Phi, \tau)$ be

$$\mathcal{L}_0(\Phi, \tau) \coloneqq \tau \cdot \log \sum_{\mu=1}^{M} \mathcal{L}_\Phi(\xi_\mu, \tau).$$

$\mathcal{L}_0(\Phi, \tau)$ converges uniformly to $\mathcal{L}_{\text{HardMax}}(\Phi)$ as $\tau \to 0$.

*Proof of Theorem 3.1.* We first organize terms in (3.2). We obtain:

$$\mathcal{L}_\Phi(\xi_\mu, \tau) = - \left[ \log \left( \exp\left\{ \frac{\langle \Phi(\xi_\mu), \Phi(\xi_\mu) \rangle}{\tau} \right\} \right) - \log \left( \sum_{\nu=1}^{M} \exp\left\{ \frac{\langle \Phi(\xi_\mu), \Phi(\xi_\nu) \rangle}{\tau} \right\} \right) \right]$$

$$= - \left[ \frac{1}{\tau} - \log \left( \sum_{\nu=1}^{M} \exp\left\{ \frac{\langle \xi_\mu, \xi_\nu \rangle}{\tau} \right\} \right) \right].$$

We define a helper function $\mathcal{L}_0$, denoted as

$$\mathcal{L}_0(\Phi, \tau) \coloneqq \tau \cdot \log \sum_{\mu=1}^{M} \ell_\mu(\Xi, \Phi, \tau). \tag{C.10}$$

We have

$$\mathcal{L}_0(\Phi, \tau) \coloneqq \tau \cdot \log \sum_{\mu=1}^{M} \mathcal{L}_\Phi(\xi_\mu, \tau)$$

$$= \tau \log \sum_{\mu=1}^{M} \log \left( 1 + \sum_{\nu \in [M] \setminus \mu} \exp\left\{ \frac{\langle \Phi(\xi_\nu), \Phi(\xi_\mu) \rangle - 1}{\tau} \right\} \right).$$

Due to the fact that $x/(1+x) \le \log(1+x) \le x$ for all $x > -1$, we have:

$$\frac{\sum_{\nu \in [M] \setminus \mu}^{M} \exp\left\{ \frac{\langle \Phi(\xi_\nu), \Phi(\xi_\mu) \rangle - 1}{\tau} \right\}}{1 + \sum_{\nu' \in [M] \setminus \mu}^{M} \exp\left\{ \frac{\langle \Phi(\xi_\nu), \Phi(\xi_\mu) \rangle - 1}{\tau} \right\}}$$

$$\le \log \left( 1 + \sum_{\nu \in [M] \setminus \mu}^{M} \exp\left\{ \frac{\langle \Phi(\xi_\nu), \Phi(\xi_\mu) \rangle - 1}{\tau} \right\} \right)$$

$$\le \sum_{\nu \in [M] \setminus \mu}^{M} \exp\left\{ \frac{\langle \Phi(\xi_\nu), \Phi(\xi_\mu) \rangle - 1}{\tau} \right\}. \tag{C.11}$$

Given the fact that $\langle \Phi(\xi_\nu), \Phi(\xi_\mu) \rangle - 1 \leq 0$ for all possible $\nu, \mu$. With the monotonicity of the exponential function, we obtain:

$$\sum_{\nu \in [M] \setminus \mu}^{M} \exp\left\{ \frac{\langle \Phi(\xi_\nu), \Phi(\xi_\mu) \rangle - 1}{\tau} \right\} \leq M - 1.$$

Combining this with LHS of (C.11), we have

$$\frac{\sum_{\nu \in [M] \setminus \mu}^{M} \exp\left\{ \frac{\langle \Phi(\xi_\nu), \Phi(\xi_\mu) \rangle - 1}{\tau} \right\}}{M}$$

$$\leq \log\left( 1 + \sum_{\nu \in [M] \setminus \mu}^{M} \exp\left\{ \frac{\langle \Phi(\xi_\nu), \Phi(\xi_\mu) \rangle - 1}{\tau} \right\} \right)$$

$$\leq \sum_{\nu \in [M] \setminus \mu}^{M} \exp\left\{ \frac{\langle \Phi(\xi_\nu), \Phi(\xi_\mu) \rangle - 1}{\tau} \right\}.$$

Summing over all possible $\mu \in [M]$ we have

$$\sum_{\mu=1}^{M} \frac{\sum_{\nu \in [M] \setminus \mu}^{M} \exp\left\{ \frac{\langle \Phi(\xi_\nu), \Phi(\xi_\mu) \rangle - 1}{\tau} \right\}}{M}$$

$$\leq \sum_{\mu=1}^{M} \log\left( 1 + \sum_{\nu \in [M] \setminus \mu}^{M} \exp\left\{ \frac{\langle \Phi(\xi_\nu), \Phi(\xi_\mu) \rangle - 1}{\tau} \right\} \right)$$

$$\leq \sum_{\mu=1}^{M} \sum_{\nu \in [M] \setminus \mu}^{M} \exp\left\{ \frac{\langle \Phi(\xi_\nu), \Phi(\xi_\mu) \rangle - 1}{\tau} \right\}.$$

Using the property of max function, we further get

$$\max_{\mu, \nu \in [M], \mu \neq \nu} \frac{\exp\left\{ \frac{\langle \Phi(\xi_\nu), \Phi(\xi_\mu) \rangle - 1}{\tau} \right\}}{M}$$

$$\leq \sum_{\mu=1}^{M} \frac{\sum_{\nu \in [M] \setminus \mu}^{M} \exp\left\{ \frac{\langle \Phi(\xi_\nu), \Phi(\xi_\mu) \rangle - 1}{\tau} \right\}}{M},$$

$$\leq \sum_{\mu=1}^{M} \log\left( 1 + \sum_{\nu \in [M] \setminus \mu}^{M} \exp\left\{ \frac{\langle \Phi(\xi_\nu), \Phi(\xi_\mu) \rangle - 1}{\tau} \right\} \right)$$

$$\leq M \cdot (M - 1) \cdot \max_{\mu, \nu \in [M], \mu \neq \nu} \exp\left\{ \frac{\langle \Phi(\xi_\nu), \Phi(\xi_\mu) \rangle - 1}{\tau} \right\}.$$

Now by taking logarithmic on both sides and multiplying all three terms by $\tau$ we get

$$\max_{\mu, \nu \in [M], \mu \neq \nu} (\langle \Phi(\xi_\nu), \Phi(\xi_\mu) \rangle - 1) - \tau \log M \leq \mathcal{L}_0(\Phi, \tau) \leq \tau \log(M \cdot (M - 1)) + \max_{\mu, \nu \in [M], \mu \neq \nu} (\langle \Phi(\xi_\nu), \Phi(\xi_\mu) \rangle - 1).$$

By $\max_{\mu,\nu\in[M],\mu\neq\nu}(\alpha_{\mu,\nu}) = \mathcal{L}_{\text{HardMax}}(\Phi) - 1$, we have

$$\mathcal{L}_{\text{HardMax}}(\Phi) - \tau\log M - 1 \leq \mathcal{L}_0(\Phi,\tau) \leq \tau\log M\cdot(M-1) + \mathcal{L}_{\text{HardMax}}(\Phi) - 1.$$

Therefore, for any $\epsilon > 0$, by taking $\tau_0 = \frac{\epsilon}{\max(\log M,\log(M\cdot(M-1)))}$, we have

$$|\mathcal{L}_0(\Phi,\tau) - \mathcal{L}_{\text{HardMax}}(\Phi)| \leq \tau\max\{\log M, \log(M\cdot(M-1))\} \leq \epsilon,$$

for any $\tau < \tau_0$. That is, $\mathcal{L}_0(\Phi,\tau)$ converges uniformly to $\mathcal{L}_{\text{HardMax}}(\Phi)$, leading to Lemma C.4.

Now we know $\mathcal{L}_0(\Phi,\tau)$ converges uniformly to $\mathcal{L}_{\text{HardMax}}(\Phi)$ as $\epsilon \to 0$, by [Rockafellar and Wets, 2009, Proposition 7.15], we have $\mathcal{L}_0(\Phi,\tau)$ $\Gamma$-converges to $\mathcal{L}_{\text{HardMax}}(\Phi)$ as well. By [Braides, 2006, Theorem 2.10], we have

$$\liminf_{\tau\to 0}\underset{\Phi\in\mathcal{H}}{\text{argmin}}\,\mathcal{L}_0(\Phi,\tau) \subseteq \underset{\Phi\in\mathcal{H}}{\text{argmin}}\,\mathcal{L}_{\text{HardMax}}(\Phi).$$

This completes the proof[8]. $\qquad\square$

## C.4 Proof of Proposition 3.1

*Proof.* We first define the one-vs-one distance.

**Definition C.1** (one-vs-one distance). We define the one-vs-one distance of a set of points $\mathcal{V} = \{v_\mu\}_{\mu=1}^M \subseteq \mathbb{S}^{d-1}$, with $|\mathcal{V}| = M$, as

$$\rho_{\text{one-vs-one}}(\mathcal{V}) := \min_{\mu\in[M]}\min_{\nu\neq\mu}\|v_\nu - v_\mu\|.$$

The one-vs-one distance is lower bounded as following

**Lemma C.5.** [Jiang et al., 2023, Lemma C.13]

$$\left[\frac{\sqrt{\pi}}{M}\frac{\Gamma\left(\frac{d+1}{2}\right)}{\Gamma\left(\frac{d}{2}+1\right)}\right]^{\frac{1}{d-1}} \leq \max_{\mathcal{V}\subseteq\mathbb{S}^{d-1}}\rho_{\text{one-vs-one}}(\mathcal{V}). \tag{C.12}$$

Note that by the definition of the one-vs-one distance, we have the equivalent expression such that

$$\frac{\rho_{\text{one-vs-one}}^2(\mathcal{V})}{2} \equiv \min_{\nu,\mu\in[M]}\langle v_\mu, v_\mu\rangle - \langle v_\mu, v_\nu\rangle.$$

Combining the above property, Lemma C.5 and a known upper bound in [Moore, 1974, Theorem 1], we obtain:

$$\frac{1}{2}\left[\frac{\sqrt{\pi}}{M}\frac{\Gamma\left(\frac{D_\Phi+1}{2}\right)}{\Gamma\left(\frac{D_\Phi}{2}+1\right)}\right]^{\frac{2}{D_\Phi-1}} \leq \Delta_{\min}^\Phi \leq 2\left[\frac{2\sqrt{\pi}}{M}\frac{\Gamma\left(\frac{D_\Phi+1}{2}\right)}{\Gamma\left(\frac{D_\Phi}{2}\right)}\right]^{\frac{1}{D_\Phi-1}}.$$

The upper bound in [Moore, 1974] is derived by the normalized surface area of a spherical cap of angular radius $\theta$. $\qquad\square$

---

[8]In general, $\mathcal{L}_0$ converges uniformly to $\mathcal{L}_{\text{hardmax}}$ as $\tau$ goes to 0 with an error rate of $|\mathcal{L}_0 - \mathcal{L}_{\text{hardmax}}| \leq 2\tau\log(M)$.

# D  Experimental Details

**Computational Environments.**    All experiments are conducted on the platform with NVIDIA GEFORCE RTX 2080 Ti and INTEL XEON SILVER 4214 @ 2.20GHz. We use PyTorch 1.8.0 for all experiments. The experiments are relatively lightweight which can also be ran on CPU-only environments.

## D.1  Metastable States

**Hyperparameters.**    The hyperparameters we used for the metastable state experiment is listed in Table 3.

Table 3: Hyperparameter used Metastable State Experiment.

| parameter | Synthetic | MNIST |
|---|---|---|
| Optimizer | Adam | Adam |
| Learning Iteration $N$ | 20 | 20 |
| Batch Size | 10 | 16 |
| Update Rule Iteration | 20 | 5 |
| Learning Rate | 0.1 | 0.1 |
| Memory set size $M$ | 10 | 60000 |
| Pattern Dimension $d$ | 5 | 784 |
| Feature Dimension $D_\Phi$ | 5 | 200 |
| $\beta$ | 4 | 0.1 |
| threshold for $p$ | 0.01 | 0.01 |

**Implementation Details.**    The batch size in Table 3 denotes the batch size we use to train the feature map $\Phi$. For the synthetic dataset, we directly train $\Phi$ on the whole memory set. For the softmax threshold, we follow the settings used in [Santos et al., 2024].

## D.2  Energy Landscape

**Hyperparameters.**    The hyperparameters we used for the basins of attraction experiment is listed in Table 5.

Table 4: Hyperparameter used in the energy landscape experiment.

| parameter | 2-Points | 4-Points |
|---|---|---|
| Optimizer | SGD | SGD |
| Learning Iteration $N$ | 5 | 5 |
| Learning Rate | 0.1 | 0.1 |
| Memory set size $M$ | 2 | 4 |
| Pattern Dimension $d$ | 2 | 2 |
| Feature Dimension $D_\Phi$ | 2 | 2 |
| $\beta$ | 20 | $0.9^{-1}$ |
| query grid resolution | $40 \times 40$ | $40 \times 40$ |
| color map | hot | hot |

**Implementation Details.**    We first prepare a set of randomly generated patterns as memories. Next we record its energy landscape with respect to different query (the coordinate in the figure). Next we train $\Phi$ for 5 iterations and record its resulting energy landscape with $N = 1, 2, 5$. We use the entmax and sparsemax package used in [Peters et al., 2019].

### D.3 Basins of Attraction

**Hyperparameters.**   The hyperparameters we used for the basins of attraction experiment is listed in Table 5.

Table 5: Hyperparameter used in the basins of attraction experiment.

| parameter | Synthetic |
|---|---|
| Optimizer | Adam |
| Learning Iteration $N$ | 5 |
| Update Rule Iteration | 5 |
| Learning Rate | 0.1 |
| Memory set size $M$ | 5 |
| Pattern Dimension $d$ | 5 |
| Feature Dimension $D_\Phi$ | 5 |
| $\beta$ | 20 |
| query grid resolution | $100 \times 100$ |

**Implementation Details.**   We specifically set the update rule iteration to 5 as we see the sharp energy gradient in Figure 1. Demonstrating that the standard MHM and its variants are not able to converge to fixed points fast.

### D.4 Simulation of Proposition 3.1

We provide a numerical simulation of our bound with $D_\Phi = 3$. We take the known solution of minimal separation published in http://neilsloane.com/packings/ a ground truth.

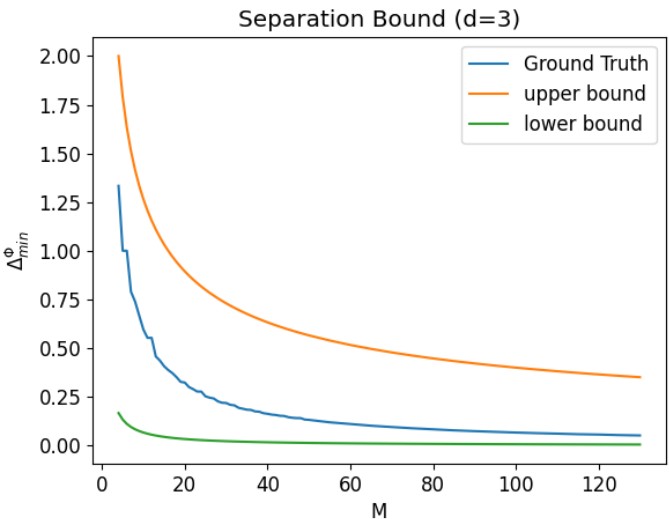

Figure 3: **Separation Bound Numerical Simulation** We visualize the bound presented in Proposition 3.1 in 3-D dimension. The bound goes tighter as the number of points increases.

### D.5 Assignment Problems

Here we conduct the point assignment problem in 2D space. In 2D space, the optimal arrangement of six points is well-established: they equally divide the unit circle, with each point neighboring two others.

We consider the case where we try to learn a feature map $\Phi$ under U-Hop+ with 6 different images sampled from CIFAR10 We sample each image from cat, dog, car, truck, deer and horse class. Intuitively, cat has closer semantic relationship with dog, car is more similar to truck and deer has closer semantic relationship with horse [Jiang et al., 2023, Neelakantan et al., 2022]. We show that our learned feature map consistently puts similar pairs closer to each other in 7 out of 10 trials. This

implies that while our method does not force or even considers the underlying semantic meanings behind each memories, our feature map is still able to present such relationship. The result is in Figure 4.

**Discussion.** The separation loss encourages the feature map to make the whole dataset the most linearly separable to each instance. A similar analysis can be found in [Dirksen et al., 2022, Ghosal et al., 2022], where they found out that if data has subtle clustered structure, a random neural network is able to make it linearly separable with high probability.

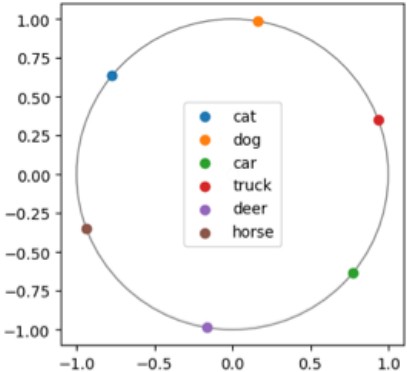

Figure 4: **Assignment Problem in 2D.** We observe that the learned feature map consistently put similar pairs closer to each other, leading to preserving some level of semantic information.

### D.6    Additional Experiments

Here we observe the loss curve of $\mathcal{L}$ w.r.t. different memory set size. We aim to verify whether $\mathcal{L}$ is able to converge well through proposed algorithm.

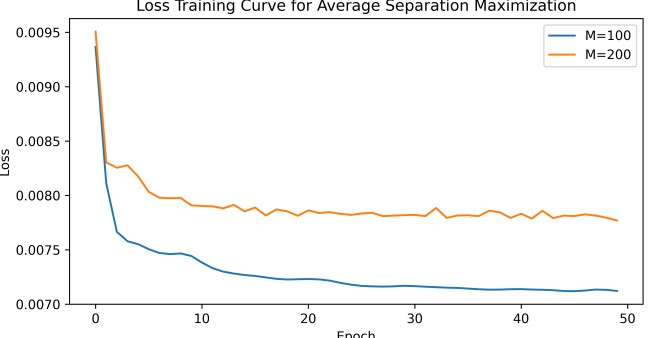

Figure 5: **Loss Curve of $\mathcal{L}$ w.r.t. different memory set size**. We run separation maximization for 100 epochs on MNIST under 2 settings, $M = 100/200$. We set $\tau = 0.1$, learning rate 1e-3, $D_\Phi = 100$. The result shows $\mathcal{L}$ converges fast, which echoes our sub-linear time complexity.

