# OpenReview forum: "Provably Optimal Memory Capacity for  Modern Hopfield Models:   Transformer-Compatible   Dense Associative Memories as Spherical Codes"
_NeurIPS.cc/2024/Conference — NeurIPS 2024 poster_

### Official Review · Reviewer_vthX · 2024-07-11

**Soundness:** 3
**Presentation:** 3
**Contribution:** 2
**Rating:** 6
**Confidence:** 4

**Summary:**

Suggest a method to optimize kernel versions of Modern Hopfield Models by mapping memories onto well-separated values on a sphere in feature space. Shows that the spherical arrangement of memories is optimal for retrieval (e.g., maximize capacity) and thus can improve upon the linear kernel [Wu et al. 2024a], which optimises memory separation.

**Strengths:**

*	Originality: the idea presented is quite simple and intuitive, but the perfected procedure is non-trivial and can potentially improve memory performance across a large family of relevant MNM-s.
*	Clarity: the submission is clearly written and well organized.

**Weaknesses:**

*	Quality: the algorithm's main justification is a lower bound that is not shown to be tight, so optimizing the minimal overlap between memories is not well supported (here).
*	Quality: if the main claim of the paper is improving the capacity to store memories, I would expect a numerical validation that the algorithm improves the number of memories that may be stored or increases the amount of noise that may be tolerated. The included results (sections 3.1 and 3.2) are only qualitative and are achieved on toy problems. Such a comparison can be done relative to plain-vanilla dense MHM or to [Wu et al. 2024a].
*	Significance: the results are mostly of practical importance for practitioners who wish to implement MHM with an optimal kernel. For that crowd, it is not that important if the algorithm is based on a novel lower bound but that the algorithm works, which needs to be better established.

**Questions:**

1.	How does the lower bound on memory capacity (lemma 2.1) contribute to the derived algorithm? It’s dependance on R and D of the embedding, but those are not optimized for in the algorithm. Furthermore, it doesn’t depend on the minimal memory separation.
2.	What is Lambda in Definition 2.8, Theorem 2.1?
3.	Is the lower bound from definition 2.7 tight? Can you establish it is not vacuous?

**Limitations:**

The authors have presented a theoretical approach based on a lower bound for capacity, which may not directly translate to improvement in task performance and has not shown such improvements through experiments.

---

> ### Author Rebuttal · Authors · 2024-08-04
>
> ### Weakness
>
> > **W1:** the algorithm's main justification is a lower bound that is not shown to be tight, so optimizing the minimal overlap between memories is not well supported (here)....:
>
> **Response:**
> Sorry for the confusion caused. We want to emphasize that the inequality in Def. 2.7 is a condition instead of a lower bound. We understand the submitted draft is a bit ambiguous about this. We have made the following modifications to clarify.
>
>
> We decompose Def.2.7 into two separate definitions:
> * >**Definition 1 (Well-Separation Condition).**
> Given a set of memory patterns
> `$\Phi(\mathcal{V}) = \{ \Phi(v_\mu)\}_{\mu=1}^M \subseteq \mathbb{S}^{D_\Phi-1}$`, we say the memory pattern $\Phi(v_\mu)$ satisfies the well-separation condition is the following holds:
> $$
> 	\Delta_{\mu}^\Phi \geq \frac{1}{\beta} \ln\( \frac{  2(M-1)}{R_\Phi}\).
> $$
>
> * > **Definition 2. (Memory Code).** Given a finite set of points `$\Phi(\mathcal{V}) = \{ \Phi(v_\mu)\}_{\mu=1}^M \subseteq \mathbb{S}^{D_\Phi-1}$`, and $\beta > 0$.
> We say $\Phi(\mathcal{V})$
> is a memory code if all points in $\mathcal{V}$ satisfies the *well-separation* condition, i.e.
> $$
> 	\Delta_{\mu}^\Phi \geq \frac{1}{\beta} \ln\( \frac{  2(M-1)}{R_\Phi}\),
> 	\quad\forall \mu \in [M].
> $$
> Further, we denote $\Lambda$ as the set of all memory codes in $\mathbb{S}^{D_\Phi - 1}$, such that
> $$
> 	\Lambda := \[ \Phi(\mathcal{V}) \quad| \quad \Phi(\mathcal{V})\quad \text{is a memory code} \].
> $$
>
> We also added intuitive explanations:
> * > Note that this inequality is a desired property of the code/data which benefits memory storage. In the following section, we demonstrate methods to achieve such property for all different $\mathcal{V}$.
>
> **Moreover, our Algorithm 1 is actually built on an upper bound, given at the inequality in `line 585`.**
>
>
> We hope these clarifications have addressed your concern.
>
> > **W2:** If the main claim of the paper is improving the capacity to store memories, I would expect a numerical validatio...
>
> **Response:**
> Thank you for pointing that out. We recognize that many reviewers have identified the same problem of lacking experiments. **In response, we have added and *multiple instance learning* and *memory retrieval*  experiments to support our main result.**
> Please see Table 2 and Figure 3 of the attached pdf.
>
> Importantly, these new experimental results indicate
> * KHMs improves the retrieval outcome by a large margin.
> * U-Hop+ improves the MHM based model in multiple instance learning tasks.
>
> > **W3:** Significance: the results are mostly of practical importance for practitioners who wish to implement MHM with an optimal kernel. For that crowd, it is not that important if the algorithm is based on a novel lower bound but that the algorithm works, which needs to be better established.
>
> **Response:**
> The main purpose of this paper is to
> 1. Identify the quantity of optimal memory capacity
> 2. Justify a theoretical grounded algorithm to achieve optimal capacity.
>
> We want to emphasize that, for practical usage, **the method provides empirical improvement with or without our justification.** We will better establish this in our experiment section. Please see attached pdf for more validations.
>
> ### Questions
>
> > **Q1:** How does the lower bound on memory capacity (lemma 2.1) contribute to ...
>
> **Response:**
> Two main reasons for us to demonstrate Lemma 2.1:
> * Showing the traditional approach in hopfield is all you need does not work under the flexibility provided by KHMs.
> * Demonstrating the exponential scaling behavior in $D_\Phi$, which corresponds to the quantity $M^\star$. To emphasize this relationship, we added the following proposition and description.
>
> >> **Proposition.** Following Theorem 2.1, let
>   $\theta \in \( 0, \frac{\pi}{2} \)$, we have
>   $$
>     	e^{ \varphi( \theta ) D_\Phi (1 + o(1)) } \
>     	\geq
>     	M^\star
>     	\geq
>     	(1 + o(1)) \sqrt{2 \pi D_\Phi} \cdot
>     	\frac{ \cos{\theta} }{ \sin^{D_\Phi-1}{ \theta} },
>   $$
>    where $\theta =  \arccos\(\rho( C_{opt} )\)$, and $\varphi(\theta) = -\log \sin{\theta}$.
>
> And a remark
>
> >> **Remark.** We have $M^\star = O(c^{D_\Phi})$, for some $c > 1$. This property echoes the exponential capacity lower bound in Lemma 2.1.
>
> ---
>
> > **Q2:** What is Lambda in Definition 2.8, Theorem 2.1?
>
> **Response:** Lambda is defined in the last line of our Definition 2.7. It refers to the set containing all memory codes.
>
> > **Q3:** Is the lower bound from definition 2.7 tight? Can you establish it is not vacuous?
>
> **Response:**
> The inequality in Definition 2.7 does not serve as a lower bound. Instead, it is a condition of data that we want to achieve with Algorithm 1 because the existence of this condition benefits memory storage. The goal of algorithm 1 is to make this condition hold for all memories in feature space.
>
> We would also like to highlight that this is a commonly used storage condition in [1, 2, 3, 4], showing that it is a well established/recognized condition.
>
> * [1] Hopfield Netorks is all you need. ICLR 2021
> * [2] On Sparse Modern Hopfield model. NeurIPS 2023
> * [3] Uniform Memory Retrieval with Larger Capacity for Modern Hopfield Models ICML 2024.
> * [4] Outlier-Efficient Hopfield Layers for Large Transformer-Based Models
>
> ===
>
> Thank you for your suggestions and attention to details!
>
> We hope that our responses adequately address your concerns, and we look forward to any further feedback.
>
> Thank you again for your time and consideration!

---

> > ### Comment · Reviewer_vthX · 2024-08-09
> > **Response to rebuttal**
> >
> > I appreciate the author's improvements in the manuscript and believe my main concerns have been addressed. I'm especially impressed by the added experiments, which, to me, provide important empirical support to the mathematical justification. I will raise my score accordingly.

---

> > > ### Author Response · Authors · 2024-08-10
> > >
> > > Dear Reviewer vthX,
> > >
> > > We are happy to hear that our revisions have addressed your concerns.
> > >
> > > Thank you again for your constructive comments, which are pivotal in improving our draft and presenting a clearer view of this work We truly appreciate your thorough review.
> > >
> > > Best,
> > >
> > > Authors

---

### Official Review · Reviewer_LSVx · 2024-07-12

**Soundness:** 4
**Presentation:** 4
**Contribution:** 4
**Rating:** 7
**Confidence:** 3

**Summary:**

This work provides a theoretical analysis of the optimal capacity on both Modern Hopfield Networks (MHNs) and Kernelized Hopfield Networks (KHNs). Specifically, it seeks to answer four fundamental problems:
1. How does memory capacity affect the gap between memories?
2. Given a fixed feature dimensionality, how to select an optimal separation function?
3. Is there a simpler and more efficient algorithm which optimize the kernel?
4. How to connect the gap between memories and the optimal capacity of KHNs theoretically?

Through the viewing of the memory set as a spherical code, the work establishes the problem of storing memories as a hard-max problem of maximizing the number of patterns stored, while ensuring that the angle between them is at least a certain constant (which the work does show). Consequently, this portrayal of the problem enables a simpler algorithm to optimize the kernel in contrast to the previous work UHop. Furthermore, the work illustrates that the optimal memory capacity is strongly dependent on the projection function $\phi$, such that its optimality maximizes the gap (or angle) between each pair of memories.

**Strengths:**

The formulation of the problem presented in the work is great while the proofs are easy to follow. Moreover, the work achieves its goals in answering its four fundamental questions while providing a nice associative memory framework, which could possibly also be applied to other MHNs.

**Weaknesses:**

There is a lack of experimentation in this work. It would be great for the work to have comparisons, similar to those detailed in [Uhop](https://arxiv.org/pdf/2404.03827).

**Questions:**

For figure 4, what is the gray dot in the center of the circle? Is it an image (of a class) in CIFAR or just a point indicating the center?

For the derivative of the RHS of the bound, $\nabla^\phi_\text{min} + \frac{1}{2\beta} \ln (\frac{1}{2} \nabla^\phi_\text{min}) \geq \frac{1}{\beta} \ln (2 (M - 1))$, is it not $\frac{1}{\beta (M - 1)}$ instead of $\frac{1}{2\beta M}$?

Is it possible to see the loss curve for MNIST when optimizing $\mathbf{W}$? It would be great to see the stability of the new $\mathcal{L}$ just to get a sense of $O(\frac{1}{-4tR^2 - \mathcal{L}^*_\phi(\Theta)})$.

**Limitations:**

There is a significant lack of experimentation (or comparisons) in this work. Other types of MHN might require modifications or a different framework to analyze their optimal capacity and compatibility with KHNs. The feature map is rather simple --- it is a linear affine function. Lastly, the spherical code framework only considers normalized points on the hypersphere.

---

> ### Author Rebuttal · Authors · 2024-08-04
>
> > **W1:**  There is a lack of experimentation in this work. It would be great for the work to have comparisons, similar to those detailed in Uhop.
>
> **Response:**
>
> Thank you for pointing that out. We recognize that many reviewers have identified the same problem of lacking experiments. **We have added additional memory retrieval and multiple instance learning experiments to support our main result.**
>
> The new multiple instance learning results and retrieval results are in Table 2 and Figure 3 of the attached pdf.
>
>
> > **Q1:** For figure 4, what is the gray dot in the center of the circle? Is it an image (of a class) in CIFAR or just a point indicating the center?
>
> **Response:**
>
> Yes, sorry for the confusion. And thanks for your attention in details! We meant to use the dot to indicate the center of the circle, it is not an image in CIFAR. We will remove it in our final version.
>
> > **Q2:** For the derivative of the RHS of the bound, …
>
> **Response:**
> Yes, you are correct. Again, thanks for your attention in details!  We have modified our proof accordingly.
>
> Meanwhile, this correction does not affect our conclusion that the RHS is increasing in M.
>
> > **Q3:** Is it possible to see the loss curve for MNIST when $W$ optimizing? It would be great to see the stability of the new $\mathcal{L}$ just to get a sense of ...?
>
> **Response:**
> Yes. Thanks for the suggestion. We provide the visualization of the loss curve for MNIST in Figure 4 of the attached pdf.
>
> ===
>
> Thank you for your suggestions and attention to details!
>
> We hope that our responses adequately address the reviewer's concerns, and we look forward to any further feedback.
>
> Thank you for your time and consideration!

---

> > ### Comment · Reviewer_LSVx · 2024-08-10
> >
> > Thank you to the authors for their detailed response and hard work. Based on the provided the results, I am happy enough to raise my score to 7.
> >
> > Side comments:
> > Yes, indeed the slight in error in RHS of the bound does not change the main statement.
> >
> > The error bars of each line in the plots provided in the response are based on standard error? If not, see if plotting the error bars as standard error could help aesthetically.
> >
> > Cheers.

---

> > > ### Author Response · Authors · 2024-08-11
> > >
> > > Thank you again for your constructive comments. We truly appreciate your thorough review!
> > >
> > > We will take the aesthetic suggestion into consideration. Thank you!

---

### Official Review · Reviewer_U6Ut · 2024-07-12

**Soundness:** 3
**Presentation:** 3
**Contribution:** 3
**Rating:** 5
**Confidence:** 5

**Summary:**

The paper is a theoretical analysis of the memory capacity of MHMs and KHMs. It establishes a connection between memory capacity and spherical codes. A method for approximating optimal memory capacity is introduced. Experiments are conducted to validate the theoretical findings.

**Strengths:**

The paper is well written and easy to follow.
The memory capacity is an important problem that can have a significant impact on SOTA architectures like Transformers.
The paper is an interesting theoretical discussion and shows a connection between memory and spherical codes in an original manner.
The theoretical framework and parts of the proofs are based on [Ramsauer et al., 2020] with adaptions to KHMs.
These previous concepts are extended to get results for optimal memory capacity and to introduce an objective for optimal memory capacity.
This objective is relaxed by an "average separation loss" introduced in [Wu et al, 2024a].

**Weaknesses:**

The optimization procedure of U-Hop is similar U-Hop+ i.e., Gradient Descent is replaced by PGD.
Due to this similarity U-Hop should have been added as baseline.
Since in the standard case MHMs use learnable weight matrices (as can be seen in Eq. 10 of [Ramsauer et al., 2020] and other publications using MHMs) there should have been
a comparison to these MHMs using gradient descent based on the average separation loss.
While memory capacity is an important property it is not immediately clear how much it helps for learning tasks. Therefore, experiments in similar vein to [Wu et al, 2024a]
should have been conducted on these tasks.

**Questions:**

How do you make sure that in Algorithm 1 the weight matrix W keeps the full column rank property from Definition 2.7?

**Limitations:**

Yes.

---

> ### Author Rebuttal · Authors · 2024-08-04
>
> > **W1:** The optimization procedure of U-Hop is similar U-Hop+ i.e., Gradient Descent is replaced by PGD. Due to this similarity U-Hop should have been added as baseline. Since in the standard case MHMs use learnable weight matrices (as can be seen in Eq. 10 of [Ramsauer et al., 2020] and other publications using MHMs) there should have been a comparison to these MHMs using gradient descent based on the average separation loss.
>
> **Response:**
> Thank you for pointing that out. The short answer to this is — **Equation 10 in [Ramsauer et al., 2020] does not ensure a well-defined retrieval update of an associative memory model.**
>
> For reasons:
> * Equation 10 will only be a legal update rule for auto-associative memory if $W_V$ is excluded, with $W_K Y, W_Q R$ being memory and query, instead of $Y, R$. The consideration of hetero-associative memory is beyond the scope of this paper.
> * Equation 10 without $W_V$ minimizes the energy function of $W_K Y$ and $W_Q R$, instead of real data $Y$, $R$. The retrieved pattern will be in $W_K Y$ space instead of $Y$ space. Thus, under MHM construction, we are not able to separate $W_K$ and $Y$.
>
> > **W2:** While memory capacity is an important property it is not immediately clear how much it helps for learning tasks. Therefore, experiments in a similar vein to [Wu et al, 2024a] should have been conducted on these tasks.
>
> **Response:**
> We recognize that many reviewers have identified the same problem of lacking experiments. In response, **we have added memory retrieval and multiple instance learning experiments to support our main results.**
>
> In our experimental result, U-Hop and U-Hop+ show similar results, but the theoretical justification of algorithm 1 makes this paper theoretically grounded.
>
> For more experimental proofs, we have added the multiple instance learning results and retrieval results in Table 2 and Figure 3 of the attached pdf.
>
>
> > **Q1:** How do you make sure that in Algorithm 1 the weight matrix W keeps the full column rank property from Definition 2.7?
>
> **Response:**
>
> In our algorithm, we do not force W to be full-rank. Existing methods such as [1, 2] are able to satisfy this property in practice. We choose to only analyze the simple PGD case to keep the convergence analysis simple. **Since $\mathbf{W}$ is in a high-dimensional space, it is almost surely full-rank** (The probability of a randomly initialized matrix having full column-rank is 1). [Remark 2.1, Wu et al 2024] also explains this implication in detail. Thus we choose to not explicitly force this property in practice.
>
> [1] McTorch, a manifold optimization library for deep learning
> [2] Trivializations for Gradient-Based Optimization on Manifolds (NIPS)
>
> ===
>
> Thank you for your time and feedback! We look forward to further discussions!

---

> > ### Comment · Reviewer_U6Ut · 2024-08-12
> >
> > Thanks to the authors for the clarifications. Thus, I will keep my score.

---

> > > ### Author Response · Authors · 2024-08-12
> > >
> > > Thank you!

---

### Official Review · Reviewer_UbwM · 2024-07-18

**Soundness:** 3
**Presentation:** 2
**Contribution:** 3
**Rating:** 5
**Confidence:** 5

**Summary:**

Modern Hopfield Networks (MHNs) are limited because they require sufficient minimal separation $\Delta_{\min}$ for theoretical guarantee. Kernelized Hopfield Models (KHNs) mitigate this limitation by storing the memories in the feature space. In particular, theoretical analysis in [Wu et al., 2024a] uses a linear kernel. The current paper proposes an algorithm to find the "optimal" feature map using spherical coding. By solving a surrogate optimization problem whose solution converges to the HardMax problem asymptotically with vanishing temperature, the algorithm improves the memory capacity.

**Strengths:**

MHNs emerge as a powerful tool for theoretical understanding of neural networks such as Transformers. The current paper builds upon KHNs and gives analysis of necessary conditions for KHNs to achieve optimal memory capacity. Further, a sub-linear time algorithm is proposed to optimize the feature map.

**Weaknesses:**

Prior work on KHNs already has analysis using linear kernel. It would be helpful to showcase the improvement brought by the current algorithm to improve clarity and contextualization relative to prior work. Moreover, the experiments only demonstrate the convergence property of the iterations. It would be nice to have more practical experiments on tasks such as multiple instance learning and retrieval. I could improve my evaluation if the authors would address the questions below.

**Questions:**

1. Since MHN is a special case of KHN, how do the current results reduce to the MHN results in [Ramsauer et al., 2020]? Are there any improvements?
2. Definition 2.7 assumes that the patterns are normalized. This assumption is not explicitly stated elsewhere and are not well-justified.
3. Definition 2.8 is more like a statement rather than a definition. Why should the definition lead to the conclusion that "there is a feature map $\Phi$ such that $\Phi(V)$ is a memory code"?
4. Is the convergence in Theorem 2.2 pointwise convergence or uniform convergence? For a given $\tau>0,$ what's the error rate?
5. The proposed algorithm learns a linear map, what about non-linear maps?
6. In line 597, since you are referring to a known upper bound in a book, it would be better to provide more details for the reader to find the result.

There are some typos:
Line 48: does -> is
Line 105: lower bounded by what?
Definition 2.5: It seems that there should be an arg in front of minmax
Definition 2.7: satisfies -> satisfy

**Limitations:**

The paper is of theoretical nature. A paragraph on limitation is given in the main content.

---

> ### Author Rebuttal · Authors · 2024-08-04
>
> > **W1:** Prior work on KHNs already has analysis using linear kernels. It would be helpful to showcase the improvement brought by the current algorithm to improve clarity and contextualization relative to prior work.
>
> **Response:**
>
> The prior work in KHN mainly explains how separation maximization helps capacity but lacks theoretical justification for why their algorithm improves capacity.
> To fill this gap, this work provides a rigorous theoretical analysis of why and how separation maximization improves KHN memory capacity. Additionally, we introduce the notion of optimal capacity and prove the asymptotic conditions for achieving it.
>
> We acknowledge that the current draft does not emphasize these points enough. We will update the final version accordingly.
>
> > **W2:** Moreover, the experiments only demonstrate the convergence property of the iterations. It would be nice to have more practical experiments on tasks such as multiple instance learning and retrieval. I could improve my evaluation if the authors would address the questions below.
>
> **Response:**  In response to more experiments, **we have added new MIL results and retrieval results** in Table 2 and Figure 3 of the attached pdf.
>
> These additional experiments demonstrate clear improvements over standard MHN.
>
>
> > **Q1:** Since MHN is a special case of KHN, how do the current results reduce to the MHN results in [Ramsauer et al., 2020]? Are there any improvements?
>
> **Response:**
>
> **The main difference of reducing to MHN is that MHN does not guarantee the well-separation condition will hold for all memories. Thus, not necessarily to store *all* the given points.**
>
> Second, MHN does not have an extra feature space to store memories. If the data dimension is too small, it is possible that the well-separation condition is impossible to achieve.
> For KHNs, since learning iteration of Algorithm 1 and $D_\Phi$ has no restrictions, one is able to select sufficiently large values of these two, to ensure the well-separation condition holds for all memories. We can also compare KHNs and MHNs with Lemma 2.1. With sufficiently large $D_\Phi$ and $N$, $R_\Phi$ will eventually surpass $R$, and thus obtain a higher lower bound.
>
> Empirically, we conducted additional exps (in Table 2, Figure 3 of the attached pdf) showing improvements of UHop+ over MHN.
>
> > **Q2:** Definition 2.7 assumes that the patterns are normalized. This assumption is not explicitly stated elsewhere and is not well-justified.
>
> **Response:**
>
> **We would like to highlight that this is a commonly used assumption in the analysis of [1, 2] and in experiments in [1, 2, 3].**
>
> A justification for this setup is its connection to transformer attention. The modern Hopfield update rule/retrieval dynamics are often compared to the attention mechanism, where the input query and memory set correspond to Q and K in attention. Since LayerNorm is commonly used in attention layers, this setup reflects real-world scenarios.
>
> Analyzing non-normalized patterns is left for future work.
>
> Lastly, we have added a highlight block for this assumption in `line 92` of the final version.
>
> > **Q3:** Definition 2.8 is more like a statement rather than a definition. Why should the definition lead to the conclusion that "there is a feature map such that is a memory code"?
>
> **Response:** Thanks for pointing this out. We recognize this potential confusion and have changed the definition to a Lemma, as it is a derived result from Definitions 2.2 and 2.7.
>
>
> > **Q4:** Is the convergence in Theorem 2.2 pointwise convergence or uniform convergence? For a given what's the error rate?
>
> **Response:**
> The main purpose of Theorem 2.2 is to show one optimization problem converges to another. The convergence between two functions here is not our main focus. That being said, the convergence between $\mathcal{L}_0$ and HardMax loss is uniform convergence as $\tau$ goes to 0. We have made it clear in our manuscript by adding the following in line 589.
>
> >> Given a $\tau > 0$, for the error rate $\epsilon$  between the convergence of $\mathcal{L}_0$ and HardMax Loss, we have $ \epsilon \leq 2 \tau \log(M)$.
>
>
> > **Q5:** The proposed algorithm learns a linear map, what about non-linear maps?
>
> **Response:**
> Yes, Algorithm 1 also works for non-linear maps.
>
> We would like to highlight that only linear feature maps have been discovered for KHNs. The explicit form of non-linear maps is not yet known. The desired (retrieval dynamics with) non-linear map should satisfy fixed-point convergence and monotonic energy minimization to be well-defined under the KHN framework. Discovering its explicit form is left for future work.
>
> > **Q6:** In line 597, since you are referring to a known upper bound in a book, it would be better to provide more details for the reader to find the result.
>
> **Response:** Thank you for pointing this out. We have modified the citation to [Thm 1, Moore 1974].
>
> > **Q7:** There are some typos: Line 48: does -> is Line 105: lower bounded by what? Definition 2.5: It seems that there should be an arg in front of minmax Definition 2.7: satisfies -> satisfy
>
> **Response:**
> Thank you very much for pointing these out!
> For Line 105, we change it to: “lower bounded by the following lemma:”
> For other typos, we have conducted another round of proofreading and fixed all typos.
>
> ===
>
> * [1] Uniform Memory Retrieval with Larger Capacity for Modern Hopfield Models. ICLR 2024
> * [2] Sparse and Structured Hopfield Networks. ICLR 2024
> * [3] Universal Hopfield Networks: A General Framework for Single-Shot Associative Memory Models
> * [4] Trivializations for gradient-based optimization on manifolds. NIPS 2019
> * [5] Vector Packing in Finite Dimensional Vector Spaces. 1974
> * [6] On kissing numbers and spherical codes in high dimensions. 2018
> * [7] New lower bounds on kissing numbers and spherical codes in high dimensions. 2023
>
> ---
> We hope these responses have addressed your concerns. Thank you!

---

> > ### Comment · Reviewer_UbwM · 2024-08-12
> >
> > Thank you for the detailed rebuttal.
> >
> > - In Fig. 3 of the attached pdf. It seems the results for "Sparse Hopfield + U-hop+" is missing.
> >
> > -
> > > "We recognize this potential confusion and have changed the definition to a Lemma, as it is a derived result from Definitions 2.2 and 2.7."
> >
> > Could you elaborate on how it is derived from Definitions 2.2 and 2.7?
> >
> > - I don't think my question Q4 is addressed.

---

> ### Author Response · Authors · 2024-08-12
>
> Thank you very much for the response.
>
> > **Q1:** In Fig. 3 of the attached...
>
> **Response:** It is almost overlapped with the Generalized Sparse Hopfield + U-hop+ line.
>
> > **Q2:** Could you elaborate on how it is derived from Definitions 2.2 and 2.7?
>
> **Response:** We apologize for the confusion. The logic behind this is that Def 2.2 leads to Def 2.7, and then Def 2.7 leads to the maximal capacity as we refer to the solution of the $\max_{ \Phi(V) } | \Phi(V) |$, and $\tilde{\Phi}$ is the solution of $\argmax_{ \Phi } \max_{ \Phi(V) } | \Phi(V) |$.
>
> > **Q3:** I don't think my question Q4 is addressed.
>
> **Response:** With the same $\Xi$ and $\Phi$, $\mathcal{L}_0$ converges uniformly to the HardMax loss with same error rate of $\epsilon = O(\tau \log M)$. However, since Thm 2.2 considers two optimization problems, it is indeed $\Gamma$-convergence between these two minimization problems (convergence for functionals). With the property of $\Gamma$-convergence, the minimizers of one minimization problem will also converges to another. We will address that in the proof of our manuscript.
>
> ____
> Thanks again for the response and your insightful review, we hope our responses address your concerns.

---

> > ### Comment · Reviewer_UbwM · 2024-08-13
> >
> > Thank you for the response. However, I think there are still some misunderstandings.
> >
> > Q1: Could you provide any intuition on why the Sparse Hopfield + U-hop+ line is almost overlapped with the Generalized Sparse Hopfield + U-hop+ line?
> >
> > Q2: In my previous question, I was wondering why Definition/Lemma 2.8 leads to the conclusion that "there is a feature map such that is a memory code". It seems to me that there are some important premises for Memory Code (Definition 2.7). One is the assumption that the patterns are normalized. Another is that the feature map $\Phi\in \mathcal{H}$ is linear.

---

> ### Author Response · Authors · 2024-08-13
>
> Glady!
>
>  > **Q1:** Could you provide any intuition on why the Sparse Hopfield + U-hop+ line is almost overlapped with the Generalized Sparse Hopfield + U-hop+ line?
>
> **Response:** Sparse Hopfield is a special case of Generalized Sparse Hopfield (GSH) with the sparsity parameter $\alpha=2$. Notably, $\alpha=2$ represents the most sparse GSH. Therefore, for datasets with high sparsity, the two may overlap.
>
>  >   **Q2:** In my previous question, I was wondering why Definition/Lemma 2.8 leads to the conclusion that "there is a feature map such that is a memory code". It seems to me that there are some important premises for Memory Code (Definition 2.7). One is the assumption that the patterns are normalized. Another is that the feature map is linear.
>
> **Response:** Thanks for pointing this out. We notice our expression causes confusion. We have changed the paragraph into the following:
>
> "From this definition, with a fixed $D_\Phi$, for some set of patterns $\mathcal{V}$ that has size $M^*$, there might exist a function $\tilde{\Phi} : \mathbb{R}^d \rightarrow \mathbb{R}^{D_\Phi}$, such that it pushes memories further from each other to the level where $\tilde{\Phi} ( \mathcal{V} )$ is a memory code.
> In other words, if one hopes to store all points in $\mathcal{V}$ with KHNs, the objective is to find an appropriate $\tilde{\Phi}$."
>
> **This paragraph is meant to be an intuitive explanation rather than a theoretical result. Thank you again for pointing this out, we have revised our manuscript accordingly.**

---

> > ### Author Response · Authors · 2024-08-13
> > **A Gentle Reminder**
> >
> > Dear Reviewer,
> >
> > As the discussion period coming to its end, we want to check if our latest response has addressed your concerns.
> >
> > We have responded to the reviewer’s followup questions and concerns. If resolved, we respectfully ask that you consider increasing your score to reflect your satisfaction.
> >
> > Please let us know if you have any further questions or need clarification. Thank you!
> >
> > Best regards,
> >
> > Authors

---

> > > ### Comment · Reviewer_UbwM · 2024-08-14
> > >
> > > Thank you for your clarifications. It appears to me that the datasets you have selected for evaluations are those with high sparsity, which makes the two curves overlap. I'm not sure such comparisons are fair. However, I think the current work makes some solid theoretical contributions. Thus, I will keep my score.

---

### Author Rebuttal · Authors · 2024-08-04

## General Response/Rebuttal Summary

Dear Reviewers,

We thank the reviewers for the insightful questions and reviews.

We have answered all the questions and addressed all the problems in detail in rebuttal and revision.

In response to the reviewers' suggestions, these revisions include additional explanations, refined definitions, paragraphs, and tables to help readers understand this paper. Most importantly, 3 new experimental studies have been added to further clarify U-Hop+'s superiority, including **Multiple Instance Learning (MIL)**, **Memory Retrieval Task** and **Loss Curve w.r.t. Different Memory Size.**

---

### **Revision Details**

**Major Revisions Include:**

* **3 Additional Experiments with Uniformly Positive Results:** [`UbwM`,`U6Ut`, `LSVx`,`vthX`]
  * **Multiple Instance Learning (MIL)**: We compare the performance of modern Hopfield and Sparse modern Hopfield with and without U-Hop+. The results show U-Hop+ constantly improves models performance in MIL tasks. [`Table 2 of Attached PDF` ]
  * **Memory Retrieval Task**: We conduct memory retrieval experiments with respect to memory size change and different noise perturbation on queries. We use MNIST and CIFAR10 as datasets. The results show with U-Hop+, retrieval dynamics enjoy lower retrieval error. [`Figure 3 of Attached PDF` ]
  * **Loss Curve w.r.t. Different Memory Size:** We plot the loss curve of Algorithm 1 on MNIST, to demonstrate the sub-linear time convergence of $\mathcal{L}_0$. [`Figure 4 of Attached PDF`]

* **Clarify Assumption of Normalized Patterns** [`UbwM, LSVx`]
  * We specify the “patterns are normalized” as a separated definition in `line 92`

* **Refined Definition 2.7:** [`UbwM`,`U6Ut`,`vthX`]
  * For clarity, we divide the original Def. 2.7 into 2 separated definitions: A & B
  * A: Well-separation condition (A condition for KHMs to storage a given memory pattern)
  * B: Memory Code (spherical codes with all points satisfy the well-separation condition)

* **A Paragraph of Comparison with Prior Works (UHop or KHN)** [`U6Ut`]
  * Prior KHN lacks theoretical justification for how separation maximization helps capacity
  * To fill this gap, this work provides a rigorous theoretical analysis of why and how separation maximization improves KHN memory capacity
  * Additionally, we introduce the notion of optimal capacity and prove the asymptotic conditions for achieving it.

* **Change Definition 2.8 to Proposition 2.1** [`UbwM`]
	* We modify Definition 2.8 into a Proposition since it is a derived result from the previous definition and theorem.

**Minor Revisions Include:**

* Proofread the manuscript and fixed all identified typos and grammatical errors by reviewers and authors.
* Added explanation for the upper bound used in `line 597`
* Added explanation for the uniform convergence between $\mathcal{L}_0$ and the HardMax loss.
* Changed `line 590` from $1/2\beta M$ to $1/ (\beta(M-1)$


We hope these revisions address the reviewers' concerns and improve the overall quality of our paper.

Thank you again for your review!

Best regards,

Authors

---

### Decision · Program_Chairs · 2024-09-25

**Decision:**

Accept (poster)

**Comment:**

There is general agreement between the reviewers that this submission is worth being accepted for its theoretical contributions. There was some debate whether empirical validations would be necessary. The authors added them during the rebuttal and they were judged diversely by the reviewers, with some being convinced, others not. Overall, I recommend acceptance as a poster and urge the authors to take extra care to incorporate the changes promised in the rebuttal when preparing the final version.